

# The role of software in science: a knowledge graph-based analysis of software mentions in PubMed Central

David Schindler[1], Felix Bensmann[2], Stefan Dietze[2,3] and Frank Krüger[1,4]

[1] Institute of Communications Engineering, University of Rostock, Rostock, Germany
[2] GESIS - Leibniz Institute for the Social Sciences, Cologne, Germany
[3] Heinrich-Heine-University, Düsseldorf, Germany
[4] Department Knowledge, Culture & Transformation, University of Rostock, Rostock, Germany

## ABSTRACT

Science across all disciplines has become increasingly data-driven, leading to additional needs with respect to software for collecting, processing and analysing data. Thus, transparency about software used as part of the scientific process is crucial to understand provenance of individual research data and insights, is a prerequisite for reproducibility and can enable macro-analysis of the evolution of scientific methods over time. However, missing rigor in software citation practices renders the automated detection and disambiguation of software mentions a challenging problem. In this work, we provide a large-scale analysis of software usage and citation practices facilitated through an unprecedented knowledge graph of software mentions and affiliated metadata generated through supervised information extraction models trained on a unique gold standard *corpus* and applied to more than 3 million scientific articles. Our information extraction approach distinguishes different types of software and mentions, disambiguates mentions and outperforms the state-of-the-art significantly, leading to the most comprehensive *corpus* of 11.8 M software mentions that are described through a knowledge graph consisting of more than 300 M triples. Our analysis provides insights into the evolution of software usage and citation patterns across various fields, ranks of journals, and impact of publications. Whereas, to the best of our knowledge, this is the most comprehensive analysis of software use and citation at the time, all data and models are shared publicly to facilitate further research into scientific use and citation of software.

# INTRODUCTION

Science across all disciplines has become increasingly data-driven, leading to additional needs with respect to software for collecting, processing and analyzing data. Hence, transparency about software used as part of the scientific process is crucial to ensure reproducibility and to understand provenance of individual research data and insights. Knowledge about the particular version or software development state is a prerequisite for reproducibility of scientific results as even minor changes to the software might impact them significantly.

Corresponding authors
David Schindler,
david.schindler@uni-rostock.de
Frank Krüger,
frank.krueger@uni-rostock.de

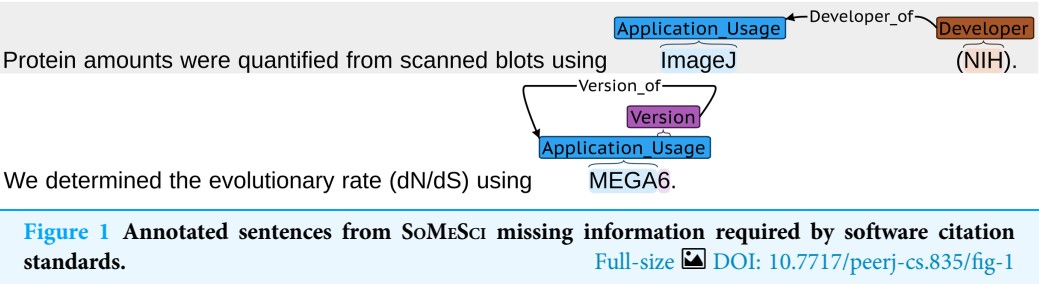

**Figure 1 Annotated sentences from SoMeSci missing information required by software citation standards.**

Furthermore, from a macro-perspective, understanding software usage, varying citation habits and their evolution over time within and across distinct disciplines can shape the understanding of the evolution of scientific disciplines, the varying influence of software on scientific impact and the emerging needs for computational support within particular disciplines and fields. Initial efforts are made to provide publicly accessible datasets that link open access articles to respective software that is used and cited, for instance, the OpenAIRE Knowledge Graph (*Manghi et al., 2019*) or SoftwareKG (*Schindler, Zapilko & Krüger, 2020*). Given the scale and heterogeneity of software citations, robust automated methods are required, able to detect and disambiguate mentions of software and related metadata.

Despite the existence of software citation principles (*Smith, Katz & Niemeyer, 2016*; *Katz et al., 2021*), software mentions in scientific articles are usually informal and often incomplete—information about the developer or the version are often missing entirely, see Fig. 1. Spelling variations and mistakes for software names, even common ones (*Schindler, Zapilko & Krüger, 2020*), increase the complexity of automatic detection and disambiguation. Training and evaluation of information extraction approaches requires reliable ground truth data of sufficient size, raising the need for manually annotated gold standard corpora of software mentions.

Most works concerned with recognition of software mentions in scientific articles apply manual analysis on small corpora in order to answer specific questions (*Howison & Bullard, 2016*; *Nangia & Katz, 2017*) or are limited to specific software (*Li, Lin & Greenberg, 2016*; *Li, Yan & Feng, 2017*). Automatic methods, enabling large scale analysis, have been implemented by iterative bootstrapping (*Pan et al., 2015*) as well as machine learning on manually engineered rules (*Duck et al., 2016*). However, both achieve only moderate performance. Extraction through deep learning with a Bi-LSTM-CRF (*Schindler, Zapilko & Krüger, 2020*) shows promise, but requires sufficient and reliable ground truth data which only recently became available.

Available corpora (*Duck et al., 2016*; *Schindler, Zapilko & Krüger, 2020*; *Du et al., 2021*) do not cover all available metadata features, cater for disambiguation of different spelling variations of the same software or distinguish between the purpose of the mention such as creation or usage. In SoMeSci (*Schindler et al., 2021b*), we have introduced a gold standard knowledge graph of software mentions in scientific articles. To the best of our

knowledge, SoMeSci is the most comprehensive gold standard *corpus* of software mentions in scientific articles, created by manually annotating 3,756 software mentions with additional information about types of software, mentions and related features, resulting in 7,237 labeled entities in 47,524 sentences from 1,367 PMC articles.

In this work, we provide a large-scale analysis of software usage and citation practices facilitated through an unprecedented knowledge graph of software mentions and affiliated metadata generated through a supervised information extraction model trained on SoMeSci and applied to more than 3 million scientific articles. In summary, our contributions include:

- **A large-scale analysis of software usage** across 3,215,386 scholarly publications covering a range of diverse fields and providing unprecedented insights into the evolution of software usage and citation patterns across various domains, distinguishing between different types of software, mentions as well as rank of journals and impact of publications. Results indicate strongly discipline-specific usage of software and an overall increase in software adoption. To the best of our knowledge, this is the most comprehensive analysis of software use and citation at the time.

- **A comprehensive knowledge graph of software citations** in scholarly publications comprising of 301,825,757 triples describing 11.8 M software mentions together with types and additional metadata. The knowledge graph is represented using established vocabularies capturing the relations between citation contexts, disambiguated software mentions and related metadata and provides a unique resource for further research into software use and citation pattern.

- **Robust supervised information extraction models for disambiguating software mentions and related knowledge** in scholarly publications. As part of our experimental evaluation, our model based on SciBERT and trained on SoMeSci *Schindler et al. (2021b)* for NER and classification outperforms state-of-the-art methods for software extraction by 5 pp on average. Software mentions are disambiguated and different variations interlinked, *e.g.*, abbreviations and name- and spelling-alternatives, of the same software.

Through these contributions, we advance the understanding of software use and citation practices across various fields and provide a significant foundation for further large-scale analysis through an unprecedented dataset as well as robust information extraction models.

The remaining paper is organized as follows. Related work is discussed in the following section, whereas the *Methods and Materials* introduces developed information extraction methods together with datasets used for training and testing. *Results: Information Extraction Performance* describes the performance results obtained on the various information extraction tasks, while the *Results: Analysis of Software Mentions* introduces an in-depth analysis of the extracted data. Key findings are discussed in the *Discussion*, followed by a brief conclusion and introduction of future work.

**Table 1 Summary of investigations concerning software in science together with source of the articles, number of articles and software, and a quality indicator.** Level of extracted details varies between listed approaches. Note that PLoS is a subset of PMC. M, manual; A, automatic; k, Cohen's; F, FScore; O, Percentage Overlap.

|   | Approach | Quality | Source | Articles | Software |
|---|----------|---------|--------|----------|----------|
| M | *Howison & Bullard (2016)* | O = 0.68–0.83 | Biology | 90 | 286 |
|   | *Nangia & Katz (2017)* | – | Nature (Journal) | 40 | 211 |
|   | *Du et al. (2021)* | O = 0.76 | PMC, Economics | 4,971 | 4,093 |
|   | *Schindler et al. (2021b)* | $\kappa$ = 0.82, F = 0.93 | PMC | 1,367 | 3,756 |
| A | *Pan et al. (2015)* | F = 0.58 | PLoS ONE | 10 K | 26 K |
|   | *Duck et al. (2016)* | F = 0.67 | PMC | 714 K | 3.9 M |
|   | *Schindler, Zapilko & Krüger (2020)* | F = 0.82 | PLoS (Social Science) | 51 K | 133 K |

# RELATED WORK

## Requirements for large scale software citation analyses

Software mentions in scientific articles have been analyzed for several reasons including mapping the landscape of available scientific software, analyses of software citation practices and measuring the impact of software in science (*Krüger & Schindler, 2020*). This includes manual analyses based on high quality data, such as *Howison & Bullard (2016)*, *Du et al. (2021)*, *Nangia & Katz (2017)* and *Schindler et al. (2021b)* but also automatic analyses such as *Pan et al. (2015)*, *Duck et al. (2016)* and *Schindler, Zapilko & Krüger (2020)*. While manual analyses provide highly reliable data, results often only provide a small excerpt and do not generalize due to small sample size. Analyses based on automatic data processing, in contrast, allow to make more general statements, for instance, regarding trends over time or across disciplines, but require high quality information extraction methods which themselves rely on reliable ground truth labels for supervised training. Table 1 compares manual and automatic approaches with respect to sample size and quality indicators such as IRR or FScore. Manual approaches provide substantial to almost perfect IRR, but are restricted to less than 5,000 articles at most. *Howison & Bullard (2016)*, for instance, analyzed software mentions in science by content analysis in 90 articles. The main objective of *Du et al. (2021)* and *Schindler et al. (2021b)* was to create annotated corpora of high quality for supervised learning of software mentions in scientific articles. *Du et al. (2021)* provide labels for software, version, developer, and URL for articles from PMC, which is multidisciplinary but strongly skewed towards Medicine (see Table A11) and Economics. *Schindler et al. (2021b)* exclusively used articles from PMC, and provide labels for software, a broad range of associated information, software type, mention type, and for disambiguation of software names.

Early automatic approaches, such as *Pan et al. (2015)* and *Duck et al. (2016)* achieve only moderate recognition performance of 0.58 and 0.67 FScore, but perform analyses on up to 714 K articles raising doubts about the reliability and generalizability of the described results. *Pan et al. (2015)* used iterative bootstrapping—a rule-based method that learns context rules—as well as a dictionary of software names based on an initial set of seed

names. *Duck et al. (2016)* employ machine learning classifiers on top of manually engineered rules. With the availability of large language models and deep learning methods for sequence labeling, *Schindler, Zapilko & Krüger (2020)* employed a Bi-LSTM-CRF and achieved an FScore of 0.82 for the recognition of software mentions in scientific articles. Most recently, *Lopez et al. (2021)* compare Bi-LSTM-CRF and SciBERT-CRF models on Softcite (*Du et al., 2021*) software entity recognition at paragraph level. They achieve 0.66 and 0.71 FScore, respectively, and further improved performance to 0.74 FScore by linking entities to Wikidata during postprocessing.

Beside high recognition rates, and thus the basis for reliable statements, *Schindler, Zapilko & Krüger (2020)* demonstrate the capabilities of semantic web technologies for information structuring and data integration with respect to analyzing software usage. They provide a KG—SoftwareKG—representing a source for structured data access for analyses. Moreover, the performed disambiguation of software mentions allows to draw conclusion on the level of software rather than software mentions, even with spelling variations. Finally, the linked nature of KG allows the integration of external data sources enabling further analyses. Following the direction of *Schindler, Zapilko & Krüger (2020)*, large scale analyses of software mentions in scholarly articles requires (1) robust information extraction and disambiguation techniques that achieve results on the level of manual approaches, and (2) the provision of all data in a standardized way that allows the reuse and the integration of external knowledge.

## Previous analyses of software in scholarly publication

As described above, previous studies on software mentions in scholarly publication were based on high quality manual analyses with small sample sizes or automatic analyses with large sample size but moderate quality. Most studies report basic descriptive statistics such as the number of overall mentions given in Table 1 or the distribution of software mentions over different software. *Howison & Bullard (2016)* report an average of 3.2 software mentions per article in Biology while *Duck et al. (2016)* report 12.9. In PMC, *Duck et al. (2016)* report an average of 5.5 mentions while *Du et al. (2021)* report 1.4 and *Schindler et al. (2021b)* 2.6. Similarly, *Pan et al. (2015)* and *Schindler, Zapilko & Krüger (2020)* report values of 2.7 and 2.6 for sub-selections of PLoS. Interestingly, *Du et al. (2021)* report a low value of 0.2 for Economics and *Duck et al. (2016)* a high value of 30.8 for Bioinformatics. Some of those results clearly show disciplinary differences, while others such as the PMC discrepancies might be attributed to methodical differences, for instance, publication time of articles in the investigated sets. Articles within *Du et al. (2021)* are significantly older than articles in *Schindler et al. (2021b)* which could result in a lower average software usage. This is also supported by the finding of *Duck et al. (2016)* who analyze software mentions up to 2013 and report a rapid increase in software usage between 2000 and 2006.

Other findings regard the distribution with respect to unique software names. *Pan et al. (2015)* report that 20% of software names account for 80% of mentions. *Duck et al. (2016)* report that 5% of software names account for 47% of mentions, and, similarly, 6.6% of entities are responsible for 50% of mentions in *Schindler et al. (2021b)*. Therefore, all prior

studies agree that the distribution of software within articles is highly skewed, pointing towards the fact that there are few pieces of general purpose software such as SPSS or R that support the scientific infrastructure. On the other hand, there is a high number of rarely mentioned software that is likely to be highly specialized towards problems and domains. *Duck et al. (2016)* perform an analysis of domain specific software to investigate disciplinary differences in software usage. They were able to confirm the existence of domain specific software and showed, for instance, that 65% of software used in medicine was not used in other analyzed domains. They also analyze journal specific software and applied a clustering analysis with respect to journal and software names.

Completeness of software mentions and citations is of high importance since employed software can only be clearly identified with sufficient information. Providing information such as the specific version or developer of software is, therefore, essential for provenance of study results or to provide credit for the creation of scientific software. For this purpose, guidelines for proper software citation have been established (*Smith, Katz & Niemeyer, 2016*; *Katz et al., 2021*) that recommend the following information to be included: name, author, version/release/date, location, venue, and unique ID, *e.g.*, DOI. *Howison & Bullard (2016)* analyze the completeness of software mentions with respect to formal citation 44%, version 28%, developer 18% and URL 5%. Based on the given information they were able to locate 86% of software online, but only 5% with the specific version. Completeness analyses by *Du et al. (2021)* showed that a total of only 44% of software mentions include further information with version being included in 27%, publisher in 31%, and URL in 17%. An analysis by *Schindler et al. (2021b)* showed that 39% mentions included a version, 23% a developer, 4% a URL and 16% a formal citation. Overall, the studies show that software mentions are still often informal and incomplete, but exhibit some notable differences between reported values. The problem of formal and informal software citation was also included in the automatic analysis of *Pan et al. (2015)* who identified formal citations for recognized software by automatic string pattern matching. They report a correlation between the number of mentions of a software and its formal citation frequency.

Availability of used software is crucial as studies conducted with commercial software might not be reproducible by other research teams. Furthermore, implementation details for non open source software cannot be reviewed by the scientific community and can potentially bias study results. Therefore, different studies included analyses regarding commercial, free and open source software usage. *Pan et al. (2015)* found that of the most frequent software mentions, which were labeled for availability manually, 64% are free for academic use. Moreover, they found that free software received more formal citations than commercial software. *Howison & Bullard (2016)* include an analysis for accessibility, license and source code availability and report that commercial software is more likely to be mentioned similar to scientific instruments (including details on developer and its location) while open access software is more often attributed with formal citations. However, they note that there is no overall preferred style for any group of software. *Schindler, Zapilko & Krüger (2020)* show a comparison of software mention

numbers for free, open source and commercial software over time that showed no clear trend towards a specific group.

Beside analyses about software in scholarly publications in general, several studies focus on particular aspects such as specific software or the relation of software usage to bibliometric measures. *Li, Lin & Greenberg (2016)*, analyze mentions of the specific engineering software (LAMMPS) and found that the given information is often not complete enough to determine how it was applied with respect to version, but also regarding software specific settings. *Li, Yan & Feng (2017)* analyze software citation for R and R packages. They report inconsistency resulting from a variety in citation standards, which are also not followed well by authors. Overall, they show a trend towards more package mentions, and find a comparably high number of formal citations for R packages (72%). *Mayernik et al. (2017)* discuss data and software citation and conclude that there is no impact measure for software available. *Allen, Teuben & Ryan (2018)* analyse the availability of source code in astrophysics and report that it could only be located for 58% of all mentions. *Pan et al. (2018)* analyze the completeness for usage statements of three specific bibliometric mapping tools and find provided versions in 30% of cases, URLs in 24%, and formal citations in 76%. They argue that the high formal citation might be due to good author citation instruction given by the tools. *Howison & Bullard (2016)* report that articles published in high impact journals mention more software. The platform swMATH (*Greuel & Sperber, 2014*) aims to establish a mapping of software used in mathematical literature by manually labeling software present in zbMATH articles pre-filtered through an automatic, heuristic search.

Most studies agree that software citation is important but often incomplete and report similar trends about the frequency of software mentions. They deviate, however, when it comes to particular numbers such as software mentions per article. This could be the result of (1) discipline specific citation habits, (2) small sample sizes in analysis studies, and (3) insufficient quality of automatic information extraction. A large scale study based on reliable automatic information extraction is required to draw conclusions across different disciplines.

# METHODS AND MATERIALS

## Information extraction

### Training dataset

We apply automatic information extraction based on supervised machine learning for recognizing software in science and use SoMeSci—Software Mentions in Science—a *corpus* of annotated software mentions in scientific articles (*Schindler et al., 2021b*). It contains 3,756 software annotations in 1,367 PubMed Central (PMC) articles as well as annotations for different software types such as *Programming Environment* or *Plug-In*, mention types such as *Usage* or *Creation*, and additional information such as *Version* or *Developer*. Moreover, it provides unique entity identities for all software annotations, which allows to not only develop a system for software name recognition but also for disambiguating

The protein-protein interaction networks were further separated into different clusters and biological significance of these clusters were

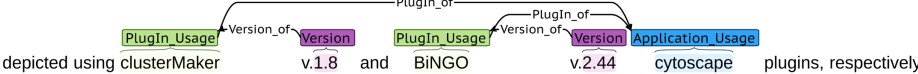

depicted using clusterMaker v.1.8 and BiNGO v.2.44 cytoscape plugins, respectively.

**Figure 2 Sentence from SoMeSci annotated with respect to software, additional information, mention type, and software type as well as corresponding relations.**

**Table 2 Overview of the SoMeSci corpus.** Further details can be found in *Schindler et al. (2021b)*.

| SoMeSci statistics | |
| --- | --- |
| # Articles | 1,367 |
| # Sentences w/ Software | 2,728 |
| # Sentences w/o Software | 44,796 |
| # Annotations | 7,237 |
| # Software | 3,756 |
| # unique Software | 883 |
| # Relations | 3,776 |
| Software Type | *Application, PlugIn, Operating System (OS), Programming Environment (PE)* |
| Mention Type | *Allusion, Usage, Creation, Deposition* |
| Additional Information | *Developer, Version, URL, Citation, Extension, Release, License, Abbreviation, Alternative Name* |

names, an essential inference step in building a software Knowledge Graph. This level of detail is not represented in other available software datasets such as BioNerDs (*Duck et al., 2016*) or Softcite (*Du et al., 2021*). SoMeSci does also contain recent articles and is, therefore, suited to represent the recent shift in awareness and recommendations for software citation. Quality of SoMeSci annotations was assessed through IRR and is reported to be high with a value of $\kappa = 0.82$. SoMeSci is available from Zenodo (https://doi.org/10.5281/zenodo.4968738) and an annotated example with markup from the web-based annotation tool BRAT (*Stenetorp et al., 2012*) is given in Fig. 2. For all reported information extraction problems described below we use the same 60:20:20 division in train, development, and test set as the SoMeSci baseline.

An overview of the different annotations along with the overall statistics of the SoMeSci dataset is given in Table 2. SoMeSci distinguishes each mention of a software by two types: mention and software. Mention type can take the values of *Usage* if the software was actively used and is contributing to the articles results, *Creation* if it was created in the scope of the article, *Deposition* if it was created and additionally published, and *Allusion* if its name was merely stated, *e.g.*, in an comparison with another software. Similarly, software type is distinguished between, *Application* if the software can be run as a stand-alone software, *PlugIn* if it is an extension to an existing host software, *Operating System* and *Programming Environment* if it is a framework for writing and executing program code. More details on the different types and relations are provided in the *Taxonomy for Software and Related Information*.

### Inference dataset

The inference dataset includes 3,215,386 articles indexed in PMC acquired *via* bulk download (https://www.ncbi.nlm.nih.gov/pmc/tools/ftp/). on January 22, 2021. Construction of SoftwareKG requires metadata and plain text of each article. To acquire the information, JATS was used instead of the also available Portable Document Format (PDF). PDF is the standard form in which humans consume scientific articles, however, there are drawbacks for machines due to formatting artifacts caused by elements such as headers, footers, page numbering, or multi-column formats. While some tools, such as *GROBID (2021)*, perform well on pdf to text conversion, using JATS prevents errors resulting from text formatting. JATS on the other hand is an XML-based format, and while specific tagging conventions vary between different journals indexed in PMC, they all follow a common scheme, making it a suited source for both metadata and plain text. Both were extracted using a custom implementation available in the associated source code (https://github.com/dave-s477/SoftwareKG).

### Entity recognition and classification

The objective of this information extraction step is to recognize software mentions and associated additional information, and to classify software according to its *Software Type* and *Mention Type*. The target labels are summarized in Table 2. The task is modelled as an NER sequence tagging problem where each sentence is considered as a sequence of tokens each of which has to be assigned a correct output tag.

Different suited state-of-the-art machine learning models are considered for the task. We compare the given baseline results on SoMeSci *Schindler et al. (2021b)*, which were established by an un-optimized Bi-LSTM-CRF model, to other machine learning models suited for scientific literature, for instance, SciBERT (*Beltagy, Lo & Cohan, 2019*). To establish a consistent naming scheme we label all implemented and tested models by *type*, classification *target* and *optimization* state: $M_{type,target,optimization}$. Results for NER are reported by mean and standard deviation for repeated training runs because performance can vary between runs due to randomization in initialization and training. Results of at least 4 different training runs are provided for hyper-parameter optimization and 16 for final performance estimation. The best model is selected on the problem of identifying software mentions ($M_{-,sw,-}$) as we consider it the most important quality measure and the main problem all other tasks relate to.

Bi-LSTM-CRFs ($M_{L,sw,-}$) were selected as they are well established for NER and have been reported to achieve state-of-the-art results (*Ma & Hovy, 2016*). Further, they have previously been applied to the problem of recognizing software in scientific literature (*Schindler, Zapilko & Krüger, 2020*; *Schindler et al., 2021b*; *Lopez et al., 2021*). More details on the model can be found in *Ma & Hovy (2016)*, *Schindler, Zapilko & Krüger (2020)* as well as in the implementation details in our published code.

BERT (*Devlin et al., 2019*) is a transformer-based model that is pre-trained on a masked language prediction task and has proven to achieve state-of-the-art performance across a wide range of NLP problems after fine-tuning. Different adaptions of the BERT pre-training procedure exist for scientific literature resulting in the two well established

**Table 3  Hyper-parameters considered for BERT models including their default setting.**

| Parameter | Default |
|---|---|
| Learning Rate (LR) | 1e−5 |
| Sampling | all data |
| Dropout | 0.1 |
| Gradient Clipping | 1.0 |

models BioBERT (*Lee et al., 2019*) ($M_{BB,sw,-}$) and SciBERT (*Beltagy, Lo & Cohan, 2019*) ($M_{SB,sw,-}$). While BioBERT is pre-trained on PubMed abstracts as well as PMC full-texts SciBERT is pre-trained on full-text articles from semantic scholar with 18% of articles coming from the domain of Computer Science and the remaining 82% from Biomedicine. To reduce run-time requirements, hyper-parameter optimization was only performed for the best performing BERT model that was chosen by comparing both models after fine-tuning with the default configuration summarized in Table 3. The parameter *Sampling* reduces the size of the training set by randomly suppressing sentences from the training *corpus* that do not contain software.

The overall, best model based on the development set is selected and extended to solve the 3 main objectives ($M_{-,sw+info,-}$) of the initial information extraction step: (1) recognize software mentions and corresponding additional information, (2) classify software type, (3) classify mention type of extracted software mentions. The combined problem is modeled as hierarchical multi-task sequence labeling and illustrated in Fig. 3. Multi-task learning can improve recognition performance and help to learn better representations if the given tasks are related as it implicitly increases the sample size (*Ruder, 2017*). Therefore, the main layers of the model share their weights across all sub-tasks and are updated with loss signals from all individual tasks. The output of each sub-task is calculated by a separate fully connected layer with softmax activation. For backpropagation we chose the simple approach of summing over the three cross-entropy losses, however, this could be further explored in the future, for instance, as described by *Kendall, Gal & Cipolla (2018)*.

The hierarchical component is added by passing the classification result of lower hierarchy sub-tasks as input to higher sub-tasks. The classification layer for mention type receives the output of software recognition and the software type layer the output of both software recognition and mention type. There is no gradient passed backward through the hierarchy so the weight updates in each classification layer are only based on the individual task loss. Teacher forcing—passing the correct prediction regardless of the actual prediction—is performed during training with respect to the output of lower layers in the hierarchy. As a result, we expect better update steps and faster learning convergence by providing more gold label information to higher classification layers. Additionally, teacher forcing should motivate the constraint that a software type or mention type should only be classified if a software was classified before. Note, that hyper-parameters for the $M_{-,sw+info,opt}$ are based on the best set of parameters identified for software recognition $M_{-,sw,opt}$.

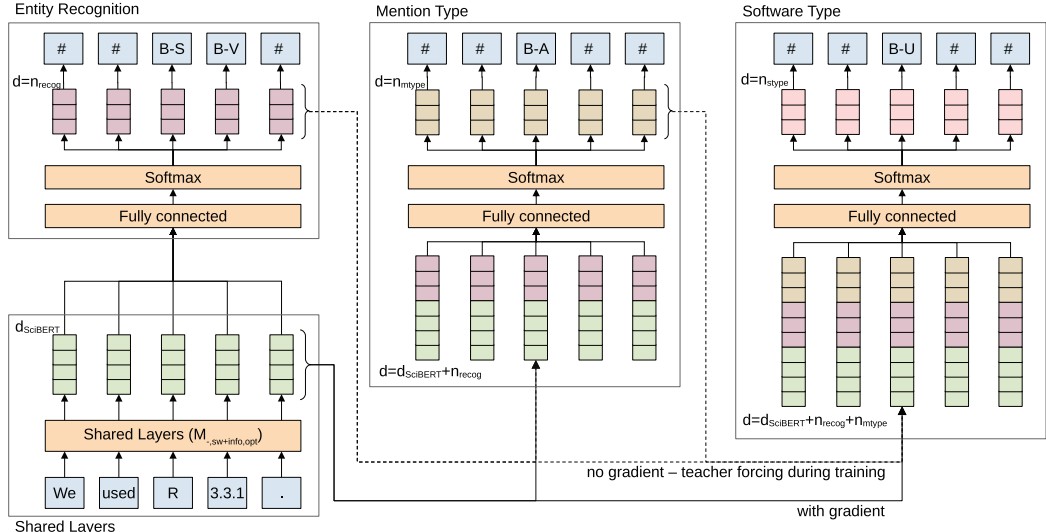

**Figure 3 Illustration of the employed multi-task, hierarchical, sequence labeling model.** Features are generated based on shared layers. The features are passed to 3 separate tasks and loss signals are summed to update shared weights. Outputs of classification layers are passed back to the network as input features to other classification layers, depicted from left to right in the image. Teacher forcing—replacing lower level classification outputs with gold label data—is used during training to stop potentially wrong classification outputs from being passed to other classification layers. Colors represent similar types of information.

**Table 4 Example for enforcing tagging consistency.** Inconsistencies are underlined.

| Sentence | We | Used | SPSS | Statistics | 16 | . |
|---|---|---|---|---|---|---|
| Entities | O | O | B-App | I-App | I-Ver | O |
| Types | O | O | B-Use | I-Mention | O | O |
| Fixed | O | O | B-App-Use | B-App-Use | B-Ver | O |

As labels for multiple tasks have to be combined with potential tagging inconsistencies for each task we experimented with adding a CRF layer on top of BERT to improve performance by learning inter-dependencies and constraints between labels. We found no improvement in performance but additional time complexity and did not further pursue the model. Instead, we enforce tagging consistencies by applying a simple set of rules: (1) all I-tags without leading B-tags are transformed to B-tags—including I-tags that do not match their leading B-tags; (2) entity boundaries for higher hierarchy tasks are adjusted to the base task entity boundaries; (3) when there are multiple conflicting labels in higher hierarchy steps for one identified software entity, the label for the first token is chosen. An example is given in Table 4.

The performance of $M_{-,sw+info,opt}$ is evaluated against the SoMeSci baseline (*Schindler et al., 2021b*) described above. In contrast to our implementation, information is not shared between tasks in the baseline model. Instead, all classifications are performed hierarchically and individually. Therefore, the reported results for the baseline are

subject to error propagation as recognition of additional information, software type classification and mention type classification all assume an underlying perfect software recognition. As our implementation does take error propagation into account the SoMeSci baseline overestimates performance in a direct comparison.

### Relation extraction

For Relation Extraction (RE), the task of classifying if and which relationships exist between entities, we considered all relations available from the training dataset. All additional information can be related to software, versions and developers to licenses, and URLs to licenses or developers. Software mentions can be related to each other by the *plugin-of* relation, representing one mention as the host software and the other as the PlugIn, or by the *specification-of* relation if both mentions refer to the same real world entity. Some possible relations are also depicted in Fig. 2. Its important to note that RE is the second information extraction step and, therefore, directly dependent on entity extraction. For developing and testing RE we rely on gold level entities, but in practice RE performance is expected to be lower due to false negatives and false positives resulting from entity extraction errors.

SoMeSci (*Schindler et al., 2021b*) provides a baseline model for classifying relations between software associated entities based on manually engineered features and an optimized Random Forest classifier. All features are implemented to yield Integer or Boolean results and take into account (1) entity order, (2) entity types, (3) entity length, (4) entity distance, (5) number of software entities, (6) sub-string relations, and (7) automatically generated acronyms.

We chose to adapt and enhance the SoMeSci baseline model instead of using more complex deep learning models because the baseline achieved good results. Moreover, RE for software associated entities is less challenging as general RE problems as we impose a large number of constraints on how entities can be related. To improve the given rule set we individually fine-tuned the implementation of each rule. Moreover, we experimented with multi-layer perceptrons and SVMs as alternative to the Random Forest classifier. In initial tests, they did not achieve better performance and we chose to retain the Random Forest classifier as it has the benefit of offering better explainability. The Random Forest was trained with 100 trees, unlimited maximum depth, and no restrictions to splitting samples.

## Software disambiguation

Software is referred to by different names due to abbreviations, geographical differences, or time. *Schindler, Zapilko & Krüger (2020)*, for instance, report up to 179 different spelling variations for the commonly used software SPSS. This raises the need for software name disambiguation as a core requirement for constructing SoftwareKG. SoMeSci provides a gold standard for this problem through manually assigned unique identifiers in form of links to external knowledge bases. However, existing knowledge bases, such as Wikidata (*Vrandečić, 2012*) or DBpedia (*Auer et al., 2007*), are sparse when it comes to

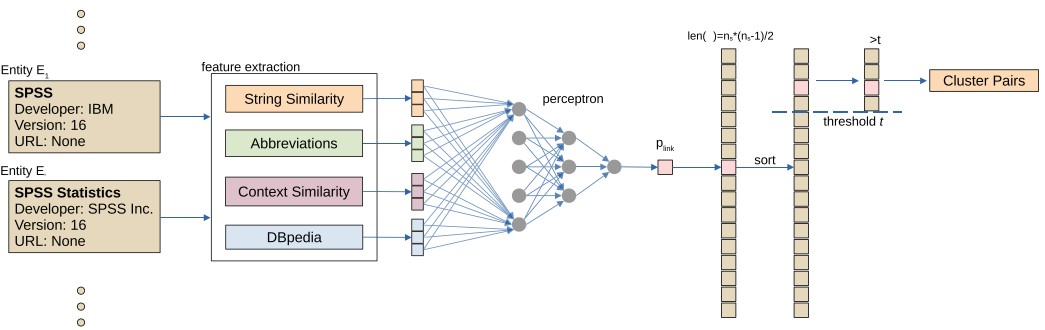

**Figure 4 Overview of the software name disambiguation. For all pairs of extracted software entities ($E_1$, $E_2$), features are extracted (feature extraction) and used to determine a probability of linking (perceptron).** Finally, agglomerative clustering is performed to cluster similar software names.

scientific software which is illustrated by an analysis of the SoMeSci disambiguation ground truth: only 205 of 883 (23%) unique and 2,228 of 3,717 (60%) software mentions are represented in Wikidata. Therefore, we adapt and develop an entity disambiguation method able to handle previously unknown software names such as those from creation statements without the need to link to external knowledge bases. In consequence, we contribute to establish a more complete KG of software.

The objective of software entity disambiguation is as follows: Given a pair ($E_1$, $E_2$) of software entities the goal is to determine whether they refer to the same real world entity. For that purpose we employ agglomerative clustering following the procedure illustrated in Fig. 4. First, manually engineered features are calculated for each pair, resulting in a feature vector $\mathbf{v}_{E1,E2}$. Features take into account: (1) string similarity, (2) similarity of extracted context information, (3) automatically generated abbreviations, and (4) software related information queried from DBpedia.

For each pair ($E_1$,$E_2$), vector $\mathbf{v}_{E1,E2}$ is mapped to a probability estimate $p_{link}$ for if they should be linked by a 4-layer perceptron ($15 \times 10 \times 5 \times 1$) with low complexity $p_{link} = f_{perceptron}(\mathbf{v}_{E1,E2})$. The model is trained supervised to predict if a link exist $l = \{0,1\}$ based on splitting all possible combinations from the ground truth set in train, development and test set in a 60:20:20 ratio. Since the class was trained as a binary classification the output of the perceptron is the result of a sigmoid layer $d \in [0,1]$ and is used in combination with a threshold in the following steps. We considered applying dropouts but found a decreasing performance in initial tests. We also did not find any increase in performance for increasing model complexity.

For disambiguation we have to consider the influence of the sample size on the density of samples in the resulting features space. For $n$ extracted mentions of software the number of entity pairs that need to be disambiguated accumulates to $n^2 - n$. In the small training set data points are less dense than in the large inference set. Moreover, the inference set does contain false positive mentions with strong resemblance to software resulting from prediction errors in the entity extraction step. This makes it difficult to find reliable decision boundaries on the training set alone. During testing it became apparent that due to the described effect the perceptron trained only on gold standard labels

could not learn suited decision boundaries to disambiguate entities pairs in the inference set. To counteract this problem, data augmentation was applied to add further entities resembling false positive extracted software names, which should not be linked to any other mentions. To simulate closeness to existing software names the new samples were generated by recombining sub-strings of existing samples, for instance, *ImageJ* and *SPSS Statistics* could be combined to form *Image Statistics*. During creation we made sure to not re-create given software names as well as duplicates. In total, $2n$ augmented samples were created once for the $n$ original software mentions and included at each training epoch. They were also included in the test set with the same factor in order to estimate performance under the chance of false positive samples. As we only add negative samples to the test set there is no risk to overestimate the performance with the employed metrics of Precision and Recall.

Based on the predicted probabilities $p_{link}$ for entity pairs a agglomerative clustering is performed. In each step, the two clusters with the largest probability are combined. As stopping criterion the threshold $t$ is introduced and defined as the minimal probability for which pairs are linked. It is optimized based on the available gold standard labels. Here, the creation of reliable decision boundaries within the densely populated feature space is also an issue. To counteract it the threshold is optimized taking into account all available data points from gold standard and inference set by combining both sets. This approach allows to evaluate how well the gold labeled mentions are clustered within the densely populated feature space. The performance is estimated in terms of Precision, Recall and FScore at $t$.

We considered single and average linkage for clustering and found almost identical performance for varying thresholds based on gold standard mentions only. Given the similar performance during the initial tests, single linkage was preferred as it offers benefits in run-time and space complexity because it allows to re-use the initially calculated similarities. Average linking, in contrast, would require additional computation for the per-cluster-pair average similarity. Due to the run-time issues described below an evaluation of average linkage would not have been feasible with the evaluation method described above. Single linkage was then applied and evaluated as described.

A major issue we faced for disambiguation was run-time complexity as the number of pairs accumulates to $n^2 - n$ with $n > 11M$ software mentions. Therefore, we had to optimize for run-time complexity. Our initial optimization step was to assume symmetric feature vectors between entities $E_1$ and $E_2$ $\mathbf{v}_{E1,E2} = \mathbf{v}_{E2,E1}$ reducing the number of required compares to $\frac{n(n-1)}{2}$, even so they are not strictly symmetric because string length of entities are included as normalization factors. Further, we made the assumption that all software with the same exact string refers to the same real-world software entity and only included a limited number of $n_{unique} = 6$ samples of each name. The work of *Schindler et al. (2021b)* showed that this can in rare cases lead to false positive clustering, but in this case the benefit outweighs this risk because otherwise the computation would not have been feasible. Disambiguation on the remaining set of ~1.4 M mentions took approximately 6 days, with feature calculations parallelized over 6 Intel® Xeon® Gold 6248 CPUs (2.50 GHz, 40 Threads).

*Schindler et al. (2021b)* provide a baseline implementation for entity disambiguation on SᴏMᴇSᴄɪ which uses manually engineered rules and external knowledge from DBpedia to disambiguate software names. For completeness we provide baseline results, however, as explained above, the density of the features space increases strongly by including additional data samples and our evaluation specifically includes augmented negative samples. Thus, the baseline cannot directly be compared to the implemented method in terms of disambiguation quality, but serves as an indicator.

## SoftwareKG: knowledge graph of software mentions
### Taxonomy for software and related information

We define software and its related information following the taxonomy presented by *Schindler et al. (2021b)* that describes the intricacies of in-text software mentions in scientific publications. The taxonomy distinguishes *Type of Software* describing which artifacts are considered as software, *Type of Mention* describing the context in which software was applied, and *Additional Information* that is provided to closer describe a software entity.

### Type of software

Based on the distinction between end-user *application* (software) and *package* introduced by *Li, Yan & Feng (2017)*, *Schindler et al. (2021b)* distinguish the following categories of software:

*Applications* are standalone programs, designed for end-users, that usually result in associated data or project files, *e.g.*, Excel sheets. This definition includes software applications that are only hosted and available through web-based services, but excludes web-based collections of data. The definition also excludes databases that are used to store collections of scientific data. To be considered as an *application* a web-service has to provide functionality beyond filtered access to data.

*Programming Environments* (PE) are environments for implementing and executing computer programs or scripts. They are built around programming languages such as Python but also integrate compilers or interpreters in order to create executables from developed code. PEs play an important role in many scientific investigations and are particularly important for computationally heavy scientific disciplines such as computer science.

*PlugIns* are extensions specifically developed to be used with existing applications or PEs and cannot be used individually. As such, in the context of PEs, the category *PlugIn* could also be called package or library. Often, the original application can be concluded from the PlugIn, *e.g.*, scikit-learn is a frequently used Python package for machine learning. The usage of Plugins is well established in the scientific community as it allows to extend the function range of well established software libraries. This allows to implement custom software without the need to establish more complex stand-alone application.

*Operating Systems* (OS) build the basis for running software on a computer by managing its hardware components and the execution of all other software. OS are necessary when running a software application and they are, overall, less mentioned than other software. In many cases authors still choose to attribute common operating systems such as Windows, OS X, or Android as well as lesser used ones such as Ubuntu or Raspbian.

## Type of mention

The definition of *Schindler et al. (2021b)* introduces a hierarchy of reasons why software is mentioned within scholarly articles based on the basic distinction between *mention* and *usage* introduced by *Howison & Bullard (2016)*:

*Allusion* of software describes each mention of a software name within a scholarly article. Aside appearance of the software name there are no further requirement for an allusion. It should especially be noted that no indication of actual usage is required, for instance, a fact about the software can be stated or multiple software can be compared. In the context of software mentions, allusions are comparable with scholarly citations used to refer to related work.

*Usage* (sub-type of Allusion) defines that a software made a contribution to a study and was actively used during the investigation, which makes the software a part of the research's provenance. Therefore, usage statements are required to allow conclusions regarding provenance. This is in line with the definition of software usage by *Lopez et al. (2021)*.

*Creation* (sub-type of Allusion) indicates that software was developed and implemented as part of a scientific investigation and is itself a research contribution. Knowledge of creation statements allows to track research software to its developers in order to provide credit to them as well as to discover and map newly published scientific software.

*Deposition* (sub-type of Creation) indicates that a software was published in the scope of a scientific investigation on top of being developed. In difference to creation statements, depositions require that authors provide either a URL to access the software or the corresponding publication license. Deposition statements, therefore, allow to provide additional information about discovered scientific software.

Both *Type of Software* and *Type of Mention* are required to fully describe a software mention in a scientific publication.

## Additional Information and Declarations

Software is constantly updated and changing. Moreover, software names are ambiguous (*Schindler, Zapilko & Krüger, 2020*). Therefore, software citation principles (*Smith, Katz & Niemeyer, 2016*; *Katz et al., 2021*) have been established to precisely identify software in publications. They require that software mentions in scholarly articles are accompanied by additional information allowing the unique identification of the actually used software, information that is often missing in practice (*Howison & Bullard, 2016*; *Du et al., 2021*; *Schindler et al., 2021b*). Here we employ the following definitions for additional information about software as defined by *Schindler et al. (2021b)*. *Developer* describes the

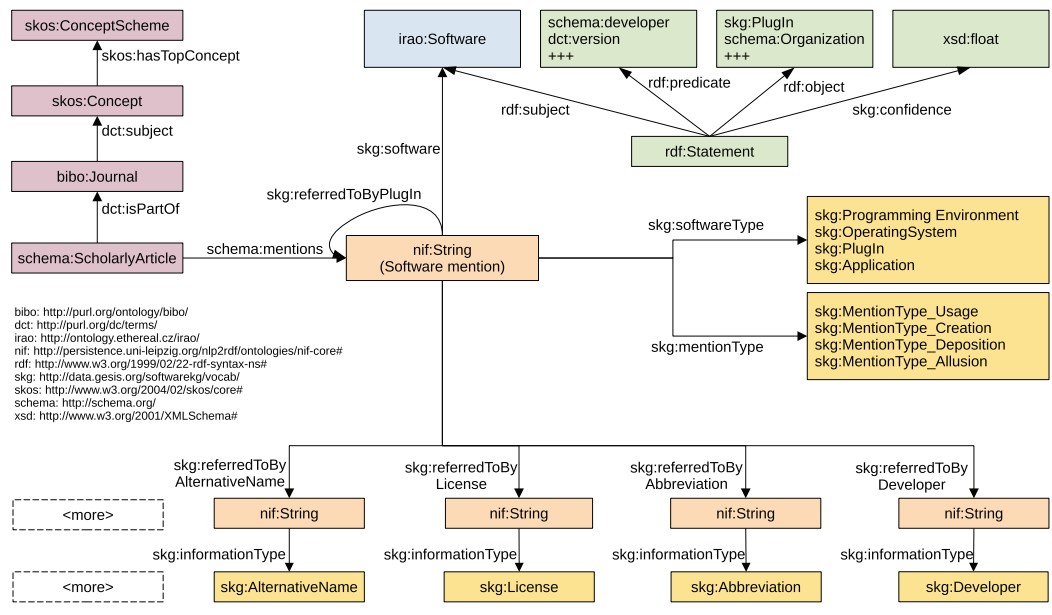

**Figure 5 Data model of the Knowledge Graph representing extracted software mentions and their related information.** For reasons of conciseness some details are left out.

person or organization that developed a software while *Version* indicates a defined state in the software life-cycle, typically identified by a version number, *Release* indicates a defined state in the software life-cycle by using a date based identifier, and *Extension* indicates different function ranges for the same base software such as professional and basic versions. *URL* gives a location for further information and download, *Citation* provides a formal, bibliographic citation, and *License* covers the permission and terms of usage. Lastly, *Abbreviation* gives a shortened name for a software while *Alternative Name* provides a longer name. All additional information is related to the specific entity it describes. In most cases this is a software, but licenses can also be specified by versions, URLs and abbreviations, while developers can be closer described by URLs and abbreviations.

### Data Model and RDF/S lifting

In order to ensure interpretability and reusability, extracted data is lifted into a structured KG based on established vocabularies. KGs represent a meaningful way to semantically structure information in an unambiguous way and provide a reasonable approach to make data accessible for later reuse. In particular, KGs enable the FAIR publication of research data.

The data model of the KG is depicted in Fig. 5. It can be subdivided into different areas that represent different types of information. Bibliographic information about articles, journal and authors (depicted in violet color) is represented by employing terms from the Bibliographic Ontology (BIBO) (*D'Arcus & Giasson, 2009*), Dublin Core Metadata Initiative Terms (DCT) (*DCMI Usage Board, 2020*), Simple Knowledge Organization System (SKOS) (*Miles et al., 2005*), and schema.org (*Guha, Brickley & Macbeth, 2016*).

The representation of entity mentions that were automatically extracted from the texts (orange color), is mainly built upon the NLP Interchange Format (NIF) (*Hellmann et al., 2013*) and Datacite (*Peroni et al., 2016*). Disambiguated software are represented by *Software* from Informatics Research Artifacts Ontology (IRAO) (*Bach, 2021*). For the metadata of software we examined dedicated vocabularies and ontologies including DOAP (Description of a Project) (*Wilder-James, 2018*), SDO (Software Description Ontology) (*Garijo et al., 2019*), SWO (Software Ontology) (*Malone et al., 2014*), OS (OntoSoft) (*Gil, Ratnakar & Garijo, 2015*), and Codemeta (*Jones et al., 2017*) (including their crosswalk), but did not use those terms as they do not represent the textual information but the real entities. For clear separation of fact and prediction we opted to not create entities from our mentions but model the mentions as they are and provide information inferred on top of them in the form of reification statements (green color). Whenever we were not able to identify existing vocabularies that allow the representation, we introduced new terms under the prefix skg (http://data.gesis.org/softwarekg/vocab/). This was necessary for modelling the information, mention and software types.

Articles and mentions are central entities of the KG. Mentions of all pieces of information extracted from an article (*schema:ScholarlyArticle*) such as software, version or developer are represented by *nif:String*. Software mentions are assigned a software type (*skg:softwareType*) and a mention type (*skg:mentionType*, yellow). For all other mentions the type is noted using the *skg:informationType* property (yellow). To represent relations at the textual level, we introduced predicates for each possible relation. The mention of a software, for instance, refers to the corresponding version *via skg: referredToByVersion*.

In order to indicate different degrees of probabilities for information aggregated over disambiguated software entities we use reification statements (*rdf:Statement*) instead of domain entities. Confidence values based on the frequency within and across articles are used to provide a measure of certainty. Formally, let $I_{r,x}$ be the set of all forms of a piece of information for a given relation $r$ and software $x$. Further, let $D$ be the set of all articles and $m_{r,a,x}$ the mapping of a piece of information $a \in I_{r,x}$ to $x$ under the relation $r$, we then define the confidence score $c_{m_{a,x}}$ as given in:

$$c_{m_{r,a,x}} = \frac{1}{|\{d \in D | m_{r,b,x} \in d, b \in I_{r,x}\}|} \cdot \sum_{d \in D} \frac{|\{m_{r,a,x} \in d\}|}{\sum_{b \in I_{r,x}} |\{m_{r,b,x} \in d\}|}, \; a \equiv b.$$

$a \equiv b$ signals that both, $a$ and $b$ represent the same type of information, *e.g.*, name or developer. This way we achieve a ratio based fair weighting on mention level and on document level. All values range from 0 to 1 and also add up to 1.

## Additional information sources

SoftwareKG was build upon data from PMC making use of the PMC OA JATS XML data set as structured information source for article metadata. Data from PubMedKG (*Xu et al., 2020*) was integrated to allow bibliometric and domain specific analyses. In particular, we used PKG2020S4 (1781-Dec. 2020), Version 4 available from http://er.tacc.utexas.edu/datasets/ped. It includes Scimago data on journal H-index, journal rank, best quartiles as

**Table 5 Development set results on software mention recognition.** Models marked with *opt* were optimized with respect to hyper-parameters, models marked with *plain* were not. Bold results highlight best performance for both plain and optimized models.

| | Precision<br>Model compare ($n = 499$) | Recall | FScore |
|---|---|---|---|
| SoMeSci Baseline | 0.82 | 0.77 | 0.79 |
| $M_{L,sw,opt}$ | 0.829 (±0.016) | 0.762 (±0.011) | 0.794 (±0.004) |
| $M_{BB,sw,plain}$ | 0.862 (±0.005) | 0.808 (±0.011) | 0.834 (±0.006) |
| $M_{SB,sw,plain}$ | **0.863** (±0.016) | **0.844** (±0.009) | **0.853** (±0.003) |
| $M_{SB,sw,opt}$ | **0.868** (±0.006) | **0.865** (±0.012) | **0.866** (±0.008) |

well as their domains and publishers. Moreover, it includes citation information for articles in PubMed from PubMed itself and Web of Science. For integration of PubMedKG we matched PMC identifiers to PubMed identifiers based on PMC's mapping service, available at https://www.ncbi.nlm.nih.gov/pmc/pmctopmid/. Specifically, we used their CSV table to match the PMC-ID in PubMed Central with PM-ID from PubMedKG.

Journal specific information vary over time so we modelled them in an *skg:JournalInfo*-entity that encapsulates information per year. Citation data are integrated in two ways: (1) all citations between PMC Open Access articles are inserted as *schema:citation* in the KG and (2) the overall number of citations an article received is included as a citation count. This allows analyses based on citation counts, but also provides a basis to identify particular citations paths.

## RESULTS: INFORMATION EXTRACTION PERFORMANCE

### Entity recognition and classification

Performance for software recognition on development set, used to select the best model, is summarized in Table 5. All values are provided by mean and standard deviation for repeated training to assess the effect of randomization in the training process of deep learning models. We found that both BERT based models perform better than $M_{L,sw,opt}$ in both Precision and Recall. As described above, $M_{SB,sw,plain}$ and $M_{BB,sw,plain}$ were initially compared with the same set of default hyper-parameters and only the best of the two models was optimized. In the initial comparison, $M_{SB,sw,plain}$ showed better performance than $M_{BB,sw,plain}$ with respect to Recall and was therefore selected. We found that hyper-parameter optimization for $M_{SB,sw,plain}$ improved performance further, especially in terms of Recall. A detailed overview of all performed hyper-parameter tests for the Bi-LSTM-CRF ($M_{L,sw,-}$) is given in supplementary Tables A1–A6 and for SciBERT ($M_{SB,sw,-}$) in supplementary Tables A7–A10. The chosen hyper-parameter configuration for $M_{SB,sw,opt}$ is summarized in Table 6. It outperforms baseline by 7 pp on the development set and is selected as the best model for the task.

The test set performance of $M_{SB,sw+info,opt}$ on all classification tasks is summarized and compared to the baseline in Table 7. Software extraction and overall entity recognition perform well with respective FScores of 0.883 (±0.005) and 0.885 (±0.005). The entity types Extension, Release, and AlternativeName, for which the fewest data samples are

**Table 6 Selected hyper-parameters for $M_{SB,sw,opt}$ fine-tuning.**

| Parameter | Value |
|---|---|
| Learning Rate (LR) | 1e–6 |
| Sampling | all data |
| Dropout | 0.2 |
| Gradient clipping | 1.0 |

**Table 7 Software mention extraction results for $M_{SB,sw+info,opt}$ in comparison with SoMeSci baseline as reported by *Schindler et al. (2021b)*, where *n* denotes the number of samples available for each classification target.** Please note that the baseline model applies hierarchical classifiers on the task and does not adjust the performance for error propagation between the initial classification of software and all other down-stream tasks. Therefore, all baseline results except for *software* are prone to overestimate performance when compared to the given results. Bold results highlight best performance in terms of FScore.

| | $M_{SB,sw+info,opt}$ | | | SoMeSci baseline | |
|---|---|---|---|---|---|
| | Precision | Recall | FScore | FScore | *n* |
| Software | 0.876 (±0.011) | 0.891 (±0.009) | **0.883 (±0.005)** | 0.83 | 590 |
| Abbreviation | 0.884 (±0.046) | 0.879 (±0.025) | **0.881 (±0.029)** | 0.71 | 17 |
| AlternativeName | 0.719 (±0.09) | 0.734 (±0.061) | **0.726 (±0.075)** | 0.25 | 4 |
| Citation | 0.868 (±0.018) | 0.855 (±0.027) | 0.861 (±0.015) | **0.87** | 120 |
| Developer | 0.867 (±0.025) | 0.901 (±0.029) | **0.883 (±0.023)** | 0.88 | 110 |
| Extension | 0.331 (±0.045) | 0.688 (±0.099) | 0.444 (±0.053) | **0.60** | 5 |
| License | 0.799 (±0.056) | 0.83 (±0.061) | **0.814 (±0.057)** | 0.80 | 14 |
| Release | 0.499 (±0.049) | 0.771 (±0.027) | 0.605 (±0.042) | **0.82** | 9 |
| URL | 0.858 (±0.028) | 0.979 (±0.006) | 0.914 (±0.016) | **0.95** | 53 |
| Version | 0.927 (±0.014) | 0.94 (±0.006) | **0.934 (±0.008)** | 0.92 | 190 |
| Entities | 0.875 (±0.009) | 0.897 (±0.009) | **0.885 (±0.005)** | 0.85 | 1,112 |
| Application | 0.788 (±0.012) | 0.865 (±0.014) | **0.824 (±0.007)** | 0.81 | 415 |
| OS | 0.933 (±0.036) | 0.852 (±0.023) | **0.89 (±0.023)** | 0.82 | 30 |
| PlugIn | 0.652 (±0.05) | 0.408 (±0.029) | **0.5 (±0.023)** | 0.43 | 78 |
| PE | 0.924 (±0.014) | 0.998 (±0.005) | 0.96 (±0.009) | **0.99** | 63 |
| Software Type | 0.792 (±0.010) | 0.818 (±0.01) | **0.800 (±0.008)** | 0.78 | 590 |
| Creation | 0.784 (±0.043) | 0.805 (±0.024) | **0.794 (±0.029)** | 0.64 | 53 |
| Deposition | 0.71 (±0.058) | 0.821 (±0.018) | **0.761 (±0.036)** | 0.65 | 28 |
| Allusion | 0.603 (±0.058) | 0.464 (±0.046) | **0.522 (±0.038)** | 0.29 | 71 |
| Usage | 0.832 (±0.013) | 0.883 (±0.011) | **0.857 (±0.007)** | 0.80 | 438 |
| Mention Type | 0.794 (±0.016) | 0.823 (±0.01) | **0.806 (±0.01)** | 0.74 | 590 |

available, show a lower performance compared to the other entities. Software types are recognized with a good overall performance of 0.800 (±0.008). Especially the types Programming Environment and Operating System are recognized with high performance. The software type Application is also recognized well, but PlugIn shows a lower performance of 0.5 (±0.023). Mention type classification also performs well with 0.806

**Table 8 Summary of RE results for both development and test set.** SOMESCI represents baseline FScores for comparison. P, Precision; R, Recall; F1, FScore; n, Number of samples per relation. Bold results highlight best performance in terms of FScore.

| | Development set | | | | | Test set | | | | |
| | Random forest | | SOMESCI | | | Random forest | | SOMESCI | | |
| Label | P | R | F1 | F1 | n | P | R | F1 | F1 | n |
|---|---|---|---|---|---|---|---|---|---|---|
| Abbreviation | 1.00 | 1.00 | 1.00 | 1.00 | 17 | 1.00 | 0.94 | 0.97 | 0.97 | 17 |
| Developer | 0.94 | 0.97 | 0.95 | 0.95 | 87 | 0.95 | 0.95 | **0.95** | 0.94 | 111 |
| AltName | 1.00 | 1.00 | **1.00** | 0.83 | 6 | 1.00 | 1.00 | 1.00 | 1.00 | 4 |
| License | 0.88 | 0.70 | **0.78** | 0.57 | 10 | 1.00 | 0.93 | **0.96** | 0.64 | 14 |
| Citation | 0.94 | 0.97 | **0.95** | 0.83 | 90 | 0.94 | 0.92 | **0.93** | 0.86 | 121 |
| Release | 0.78 | 1.00 | **0.88** | 0.80 | 7 | 0.88 | 0.78 | **0.82** | 0.53 | 9 |
| URL | 0.93 | 0.94 | **0.94** | 0.80 | 70 | 0.98 | 0.92 | **0.95** | 0.89 | 53 |
| Version | 0.97 | 0.99 | **0.98** | 0.96 | 139 | 0.98 | 0.96 | **0.97** | 0.95 | 190 |
| Extension | 1.00 | 1.00 | 1.00 | 1.00 | 5 | 1.00 | 1.00 | **1.00** | 0.89 | 5 |
| PlugIn | 0.77 | 0.66 | **0.71** | 0.60 | 35 | 0.85 | 0.72 | **0.78** | 0.65 | 39 |
| Specification | 0.67 | 0.67 | **0.67** | 0.60 | 6 | 0.83 | 0.62 | **0.71** | 0.22 | 8 |
| Overall | 0.93 | 0.94 | **0.93** | 0.87 | 472 | 0.95 | 0.92 | **0.94** | 0.88 | 571 |

(±0.01). Here, mention type Allusion is the most challenging target with 0.522 (±0.038) FScore, while all other targets are extracted with a satisfactory performance.

## Relation extraction

The results for RE by the Random Forest as well as the original SOMESCI baseline on both, development set and test set are given and compared in Table 8. In summary, high recognition rates with 0.94 FScore are observed, showing improvements resulting from our extension and optimization to the baseline. At the level of the individual relation types, high FScores (>0.9) are observed for all types except for Release $F1 = 0.82$, PlugIn $F1 = 0.78$, and Specification $F1 = 0.71$. This indicates that relations between two software entities, *i.e.*, PlugIn and Specification, are particularly challenging classification targets.

## Software disambiguation

As described above, the disambiguation first uses a perceptron model to estimate linking probabilities between entity pairs and afterwards uses the probabilities for agglomerative clustering. The optimized perceptron predicted links between software entities with a Precision 0.96, Recall 0.90, and FScore 0.93. These values were estimated on the test dataset with a threshold of $t = 0.5$ based on the sigmoid output. The perceptron performance does influence the final performance, but during clustering entity pairs $p_{link}(E_A,E_C) < t$ can still be linked even if they have not been predicted by the perceptron through a chain of closer entities: $p_{link}(E_A,E_B) > t$, $p_{link}(E_B,E_C) > t$. Therefore, evaluation of the perceptron alone does not allow to make statements about disambiguation quality.

For the actual agglomerative clustering based on single linkage performance was estimated with a Precision of 0.99, Recall of 0.96, and FScore of 0.97 at a optimal threshold of $t = 0.00347$ for clustering all gold label data in a common features space with all

extracted data. The SoMeSci baseline results are reported with Precision of 0.99, Recall of 0.96 and FScore of 0.97, but as noted above they cannot be compared to the results reported here. The small threshold is a clear indicator of how densely populated the feature space is considering all extracted software mentions. In total 605.364 clusters of software were generated.

## RESULTS: ANALYSIS OF SOFTWARE MENTIONS

### KG statistics

The KG was constructed from 301,825,757 subject-predicate-object triples, representing 11.8 M software mentions in more than 3.2 M articles in 15,338 journals from 2,136 publishers. On average, each journal contains 210 articles, ranging from 1 article to 239,962 articles in the journal PLoS One.

For ~8.7 K journals (containing 2.8 M articles, 86.7%) additional information, including citations, research domain and journal ranks was identified from integrating data of PubMedKG (*Xu et al., 2020*). For almost 2.5 M articles a citation count different from 0 could be found. In summary, 303 categories from 27 top level domains were found, see Table 9.

A detailed overview of article and journal frequencies per research domain is provided in supplementary Table A11. As expected from a repository of Open Access articles from Biomedicine and Life Sciences, the distribution of journals and articles is skewed towards Medicine, as ~1.9 M articles from 4,455 journals are related to Medicine, while only 2,178 articles from 181 journals are related to Economics. However, there is a high relative amount of articles not directly related to medicine (more than 30%). This includes disciplines such as Computer Science (~77 K articles from 396 journals) and Mathematics (~39 K articles 364 journals), but also Business (~3 K articles from 173 journals) and Arts and Humanities (~9 K articles from 469 journals).

### Software mentions

Different spellings of the same software were grouped during disambiguation, resulting in 605,362 unique software instances with 1.08 different spellings and 19.48 mentions on average. A highly skewed distribution of mentions per software can be observed, where about 10% of the software account for about 90% of the software mentions across all articles. Figure 6 illustrates this distribution graphically. Table 10 provides an overview of the 10 most frequent software, including their absolute and relative number of mentions across all articles. Furthermore, the number of articles containing at least one mention of the respective software is given in the column # Articles. With 539,250 respectively 469,751 mentions, SPSS and R are mentioned most frequently across all articles, where 440 different spellings were observed for SPSS and only 1 for R. The different spellings for SPSS include common names such as "SPSS" (78.4%), "SPSS Statistics" (10.8%), and "Statistical Package for the Social Sciences" (3.8%), but also those with spelling mistakes such as "Statistical Package for the Spcial [*sic*] Sciences".

Figure 7 illustrates the top 10 software per research domain. Domain-specific differences can be observed from the plot. No domain is consistent with the

**Table 9 Overview of the 27 main research domains and 3 of their sub categories that were used to group journals.** Bold font highlights the abbreviation of the respective research domain used here.

| Main research domain | Research subcategories (excerpt) |
| --- | --- |
| **Physics** and Astronomy | Acoustics and Ultrasonics, Astronomy and Astrophysics, Atomic and Molecular Physics, and Optics |
| **Chemistry** | Analytical Chemistry, Chemistry (miscellaneous), Electrochemistry |
| **Social** Sciences | Anthropology, Archeology, Communication |
| **Materials** Science | Biomaterials, Ceramics and Composites, Electronic |
| **Engineering** | Aerospace Engineering, Architecture, Automotive Engineering |
| **Economics**, Econometrics and Finance | Economics and Econometrics, Economics, Econometrics and Finance (miscellaneous) |
| **Multidisciplinary** | Multidisciplinary |
| **Energy** | Energy (miscellaneous), Energy Engineering and Power Technology, Fuel Technology |
| **Agricultural** and Biological Sciences | Agricultural and Biological Sciences (miscellaneous), Agronomy and Crop Science, Animal Science and Zoology |
| **Environmental** Science | Ecological Modeling, Ecology, Environmental Chemistry |
| **Veterinary** | Equine, Food Animals, Small Animals |
| **Nursing** | Advanced and Specialized Nursing, Assessment and Diagnosis, Care Planning |
| **Decision** Sciences | Statistics, Probability and Uncertainty, Information Systems and Management |
| **Earth** and Planetary Sciences | Atmospheric Science, Computers in Earth Sciences, Earth and Planetary Sciences (miscellaneous) |
| **Pharmacology**, Toxicology and Pharmaceutics | Drug Discovery, Pharmaceutical Science, Pharmacology |
| **Mathematics** | Algebra and Number Theory, Analysis, Applied Mathematics |
| **Computer** Science | Artificial Intelligence, Computational Theory and Mathematics, Computer Graphics and Computer-Aided Design |
| **Biochemistry**, Genetics and Molecular Biology | Aging, Biochemistry, Biochemistry |
| **Dentistry** | Dentistry (miscellaneous), Oral Surgery, Orthodontics |
| **Neuroscience** | Behavioral Neuroscience, Biological Psychiatry, Cellular and Molecular Neuroscience |
| **Arts** and Humanities | Archeology (arts and humanities), Arts and Humanities (miscellaneous), Conservation |
| **Psychology** | Applied Psychology, Clinical Psychology, Developmental and Educational Psychology |
| **Business**, Management and Accounting | Accounting, Business and International Management, Business |
| **Medicine** | Anatomy, Anesthesiology and Pain Medicine, Biochemistry (medical) |
| **Immunology** and Microbiology | Applied Microbiology and Biotechnology, Immunology, Immunology and Microbiology (miscellaneous) |
| **Health** Professions | Chiropractics, Complementary and Manual Therapy, Health Information Management |
| **Chemical** Engineering | Bioengineering, Catalysis, Chemical Engineering (miscellaneous) |

domain-independent view (see Table 10), and each domain is characterized by a different distribution of the top 10 software. While SPSS (top 1 for 13/27) and R (top 1 for 6/27) together represent the top mentioned software in more the 70% of the domains, Excel, BLAST, Prism, and ArcGIS (each 1/27) are the top software in Economics, Energy, Immunology, and Business, respectively. The software SHELXL, SHELXS, SAINT, and SHELXTL play a mayor role in Chemistry, Materials, and Physics, taking ranks among 1–5 in each of these domains, but are not among the top 10 in any other field. Several programming environments are listed among the top 10 software, including R, Python, Java, C, and C++, most prominent in Mathematics, Engineering, and Computer Science. Material Science plays a special role, when it comes to domain specific software because

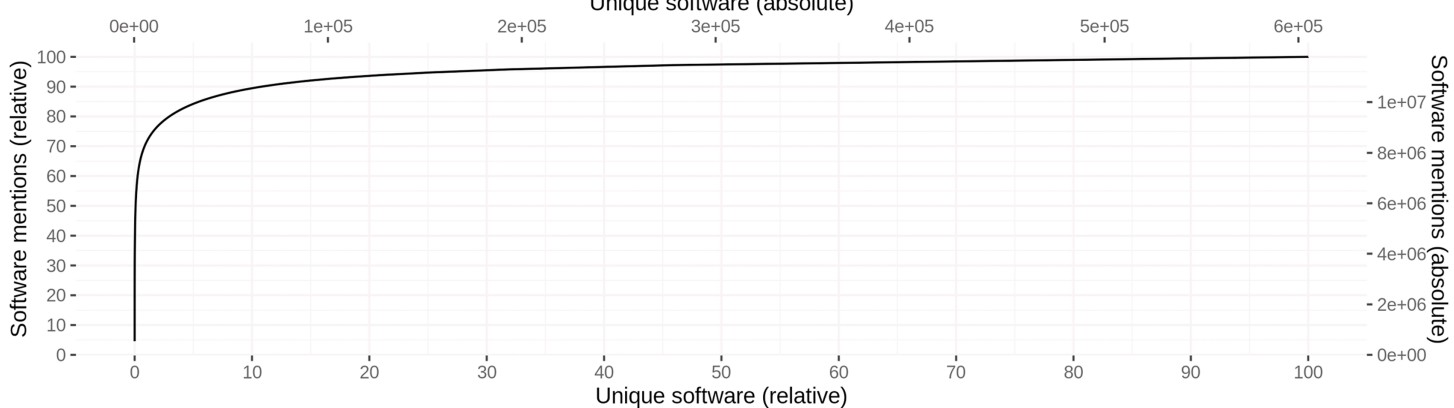

**Figure 6 Cumulative distribution of software mentions per unique software.** Left (bottom) scale gives the relative values, whereas right (top) scale provides the absolute numbers.                                                            

**Table 10 Information about the 10 most frequent software mentions across all disciplines together with their absolute and relative number of mentions, the number of articles that contain at least one mention and the number of spelling variation that could be disambiguated.**

| Software | Absolute # | Relative # | # Articles | # Spellings |
|----------|-----------|-----------|-----------|-------------|
| SPSS | 539,250 | 4.57 | 466,505 | 440 |
| R | 469,751 | 3.98 | 235,180 | 1 |
| Prism | 220,175 | 1.87 | 189,578 | 1 |
| ImageJ | 228,140 | 1.93 | 144,737 | 83 |
| Windows | 140,941 | 1.19 | 127,691 | 6 |
| Stata | 147,586 | 1.25 | 118,413 | 141 |
| Excel | 151,613 | 1.29 | 118,082 | 54 |
| SAS | 140,214 | 1.19 | 112,679 | 215 |
| BLAST | 271,343 | 2.30 | 104,734 | 383 |
| MATLAB | 160,164 | 1.36 | 89,346 | 6 |

the Operating System Windows is the only software that is shared with the top 10 overall software, while the remaining software allows a unique characterization. Regarding operating systems, Windows is frequently mentioned in all research domains, except for Earth and Planetary Sciences and Energy. The Linux operating system, in contrast, is ranked 5th in Mathematics, 7th in Decision Sciences, and 9th in Computer Science, respectively. The source code and data repositories GitHub and FigShare are listed among the top 10 software in Decision Science with rank 5 and 9.

## Article level statistics

On the article level, we observe that each article contains 3.67 software mentions on average, ranging from 0 software mentions for 1,301,192 articles to a maximum of 673 mentions for one article. Looking at the number of articles per year, it can be observed that the relative number of articles mentioning at least one software increases over all articles. Figure 8 (blue line) illustrates this graphically. Considering those articles only, a similar

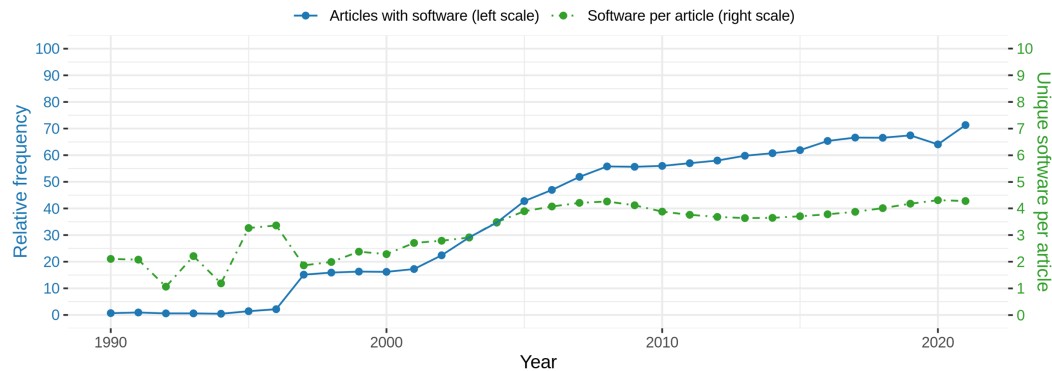

**Figure 7 Top 10 software per domain.** Higher rank within the domain is represented by darker color. The number on the tile gives the rank within the domain. Software with rank higher than 10, are excluded from the plot to improve readability. Software are ordered by rank over all domains left to right.

**Figure 8 Blue: Relative frequency of articles with at least one software mention per year. Green: Absolute mean frequency of unique software mentioned per article with at least one software mention per year.** Please note that standard deviations are at the same level as the actual average values but are omitted here for reasons of readability.

trend can be observed from the mean frequency of software mentioned within one article (Fig. 8, green line). Due to the low number of software mentions before 1997 (blue line), the estimation of mean software frequencies per article is less reliable before 1997.

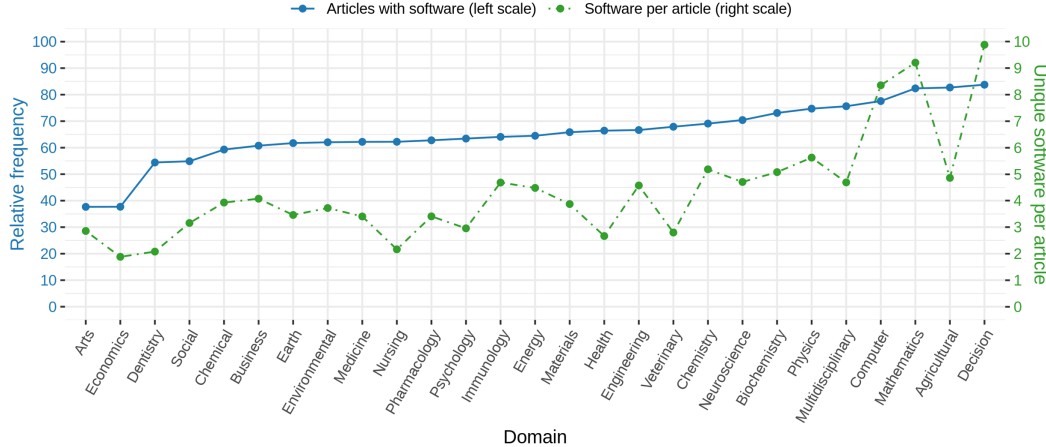

**Figure 9** **Blue: Relative frequency of articles with at least one software mention per research domain. Green: Average number of different software mentioned per article with at least one software mention given by research domain.** Note that standard deviations are large (similar to average values) and are omitted here.

In 1997, a steep increase in the number of articles with software can be observed which remains constant until 2000. From 2001 another increase until 2008 can be observed, which is followed by a phase where the relative frequency of articles with software increases more slowly until 2021. From 2007 more than 50% of the articles contain at least one software, increasing to almost 75% in 2021. A notable decrease was observed for 2020. With respect to the number of software per article (green line), the frequency remains at a constant level of ~4 from 2005. Standard deviations are omitted but are on a high level between 2 for low mean values and 4 for higher means. To determine the effect of the year on the number of software per article, a linear model was fitted to explain the binary logarithm of the number of software per article by the interaction of year and domain. We found a significant ($p < 0.001$) but small influence of the year (slope = 0.017, $S_e = 0.0006$, $R^2$ = 0.06525), when considering the interaction with the domain.

When looking at the relative amount of articles with software per research domain, we found notable differences between the individual domains. Figure 9 (blue line) illustrates those differences graphically. While in Arts and Economics only 40% of the articles contain software mentions, in Mathematics, Agriculture and Biological Sciences, and Decision Sciences, more than 80% of the articles mention at least one software. The number of different software per article draws a similar picture (Fig. 9, green line), ranging from values of 2 or 3 in Arts, Economics, and Dentistry to values above 8 for Computer Science, Mathematics, and Decision Science. A one-way ANOVA revealed these differences to be significant ($p < 0.001$, $F_{26,3468692}$ = 7950.1).

Comparing the amount of articles with software mentions with journal rank and citation count per year, similar observations were found. Both graphs are illustrated in Fig. 10. The graph illustrates the ventiles (20-quantiles) of the journal rank, grouped by domain to prevent domain specific biases due to higher journal ranks. In detail, in the first step, for each domain, the journals were distributed according to their rank ventiles and the resulting ventiles were then merged across all domains. A similar approach was

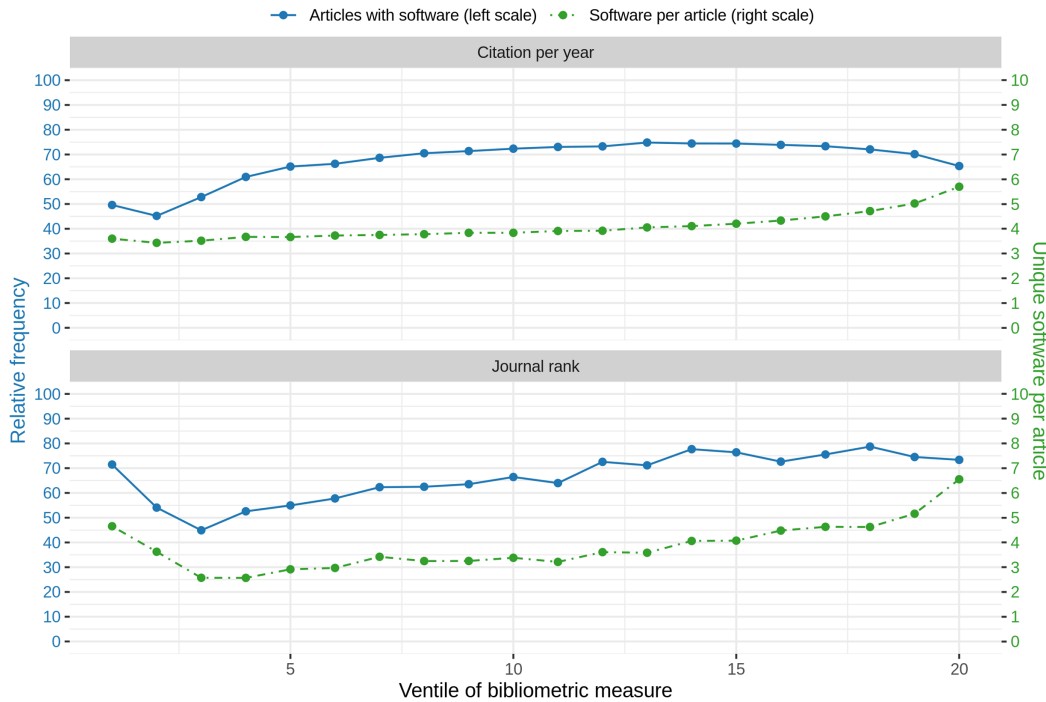

**Figure 10 Blue: Relative frequency of articles that contain at least one software mention per rank of bibliometric measure. Green: Average number of different software per article per bibliometric measure.** Note that the high standard deviation (at the same level as average values) are left out to increase readability.

chosen for summarizing the articles *via* citation count ventiles. Please note that while for journal rank based analysis all ranked journals could be considered, for citation count analysis we restricted the analysis to all articles published before 2020 to reduce a bias towards 0 citations. When considering the journal rank, we found that almost 75% of the articles on the lowest rank contain software mentions, followed by a strong decrease for the next two ventiles (blue line). For the remaining ventiles an increasing trend up to almost 80% for higher journal ranks could be observed. When considering the amount of software per article, an initial high-point and decrease for the four lower journal ranks could be observed followed by an increasing trend with increasing journal rank (Fig. 10, green line). A linear model to explain the relation between binary logarithm of the number of software per article and the journal rank grouped by domain revealed a small but significant ($p < 0.001$) effect (slope = 0.18, $S_e = 0.0017$, $R^2 = 0.087$).

Similarly, a high value followed by a slight decrease could be observed for the relative number of articles mentioning at least one software per citation count per year (blue line), even though the pattern is not that distinctive. After reaching the minimum frequency of articles containing software mentions in the 2nd ventile, the graph shows an increasing trend for the remaining ventiles, with a slow decrease for the last seven ventiles. The maximum frequency of articles with software mentions is reached at 13th citation count ventile. A linear model to explain the relation between binary logarithm of the

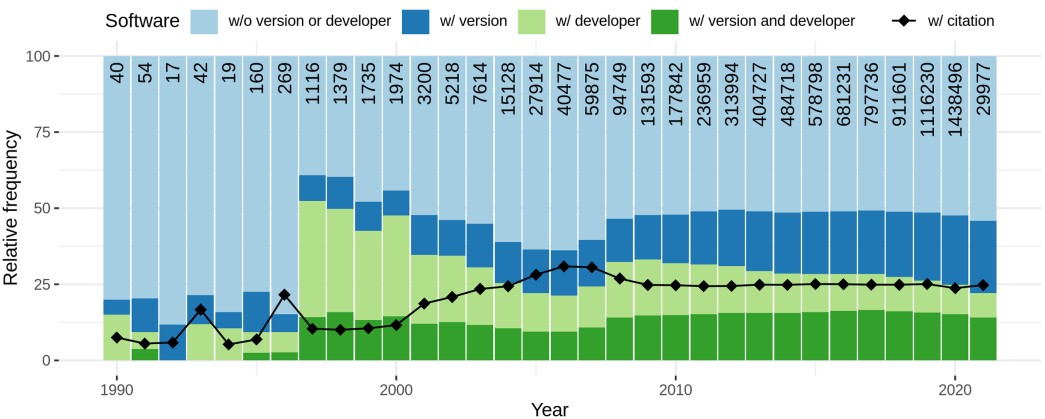

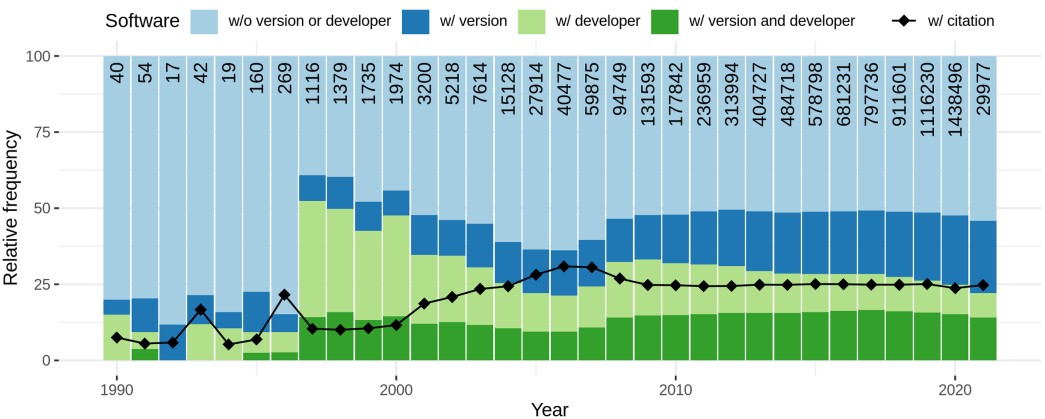

**Figure 11 Distribution of software completeness per year with the percentage of unique software per article that is cited with provided additional information.** The colored bars represent the different levels of completeness while the line chart separately indicates how many software mentions were accompanied by a formal citation. The numbers at the top of the bars represent the absolute number of software considered per year.

number of software per article and the citation count per year grouped by domain revealed a small but significant ($p < 0.001$) effect (slope = 0.01, $S_e = 0.0003$, $R^2 = 0.065$).

## Software citation completeness

Considering articles containing software mentions, we analysed the amount of information necessary to identify particular software provided for each software mentioned within one article. For each unique software (which might be referred to multiple times within the same article) we examined whether the version and/or the developer was mentioned within the article. Moreover, the frequency of formal citation (a citation referring to the bibliography of the article) was investigated.

Figure 11 depicts the completeness of software mentions per year. From the numbers at the top of the bars, it can be seen that the number of unique software per article increased over the years, ending with 1.44 M software mentions (unique per article) in 2020. The low number in 2021 reflects the time of data retrieval. Regarding citation completeness, from 1990 to 1996 the amount of both, information accompanying mentions as well as formal citations, is low for an overall low number of software mentions. From 1997 to 2000 there is a peak in additional information provided for software mentions with a low number of corresponding formal citations. Afterwards, up to year 2007, there is a decay in additional information for informal mentions and a contrary increase in formal citations. From 2008 to 2010 there is another increase of the amount of provided information and a decrease in formal citation. The numbers then stagnate up to 2021. Overall, it can be seen that the frequency of software with developer decreases and with version increases. Both, the relative amount of formal citations and the amount of software accompanied with version and developer remained constant since 2009.

Considering domain specific software citation habits, Fig. 12 illustrates the amount of information provided per software over different research domains. Mathematics, Decision Science, and Computer Science provide the least additional information with

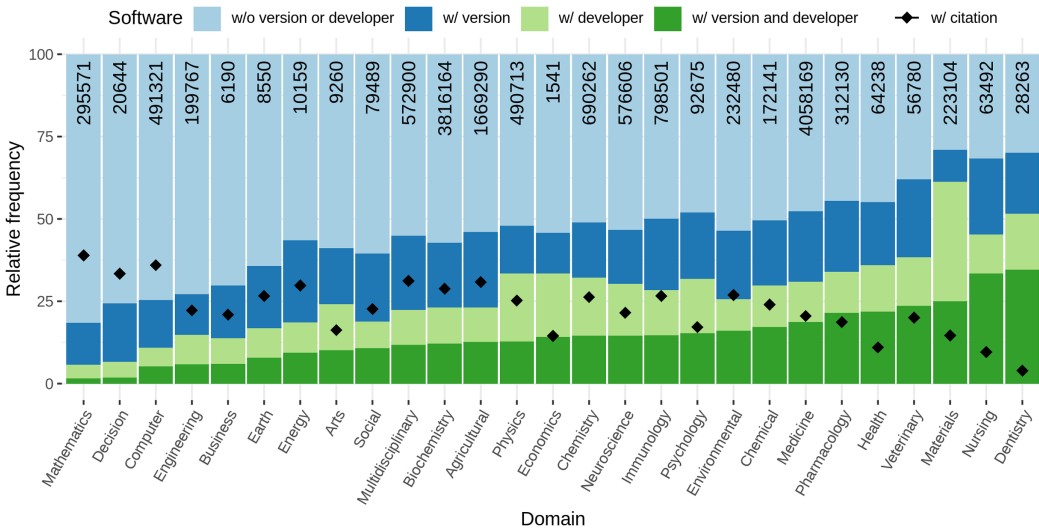

**Figure 12 Distribution of software completeness per research domain.** The numbers at the top of the bars represent the absolute numbers of software considered per domain. Please note that articles may belong to multiple categories.               

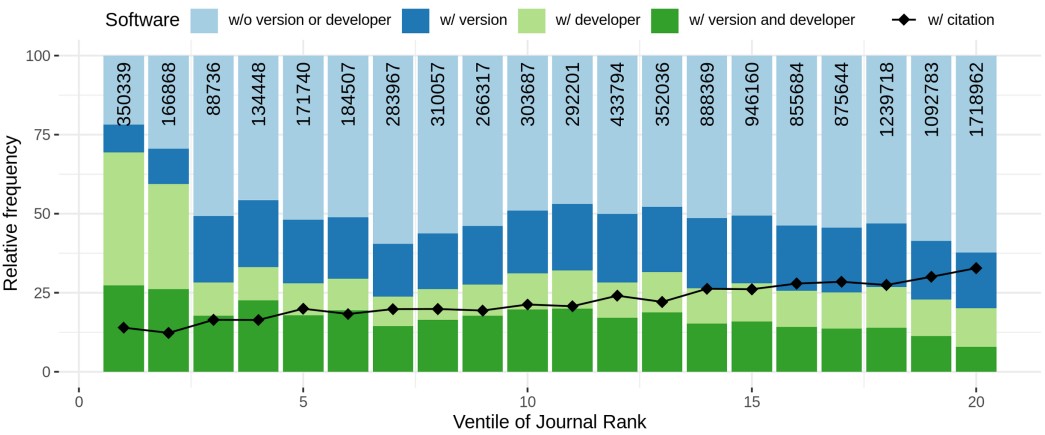

**Figure 13 Distribution of software completeness per ventile of journal rank per research domains.** The numbers at the top of the bars represent the absolute numbers considered per ventiles.

down to 20% of mentions, but all three have comparably high numbers of formal citation with around 40%. Dentistry, Nursing, and Materials, in contrast, provide most additional information with up to 70% of mentions but low numbers of formal citations with down to 5%. Nursing and Dentistry also have the highest share of software together with version and developer. In general, citation completeness is not better in domains that use most software.

With respect to journal rank, a slight negative correlation between the rank ventile and the amount of additional information can be observed. Figure 13 illustrates this relation graphically. In contrast, a positive correlation between amount of formal citation and journal rank can be seen. While most software are accompanied with developer for low ranked journals (almost 70%), the percentage decreases with rising journal rank,

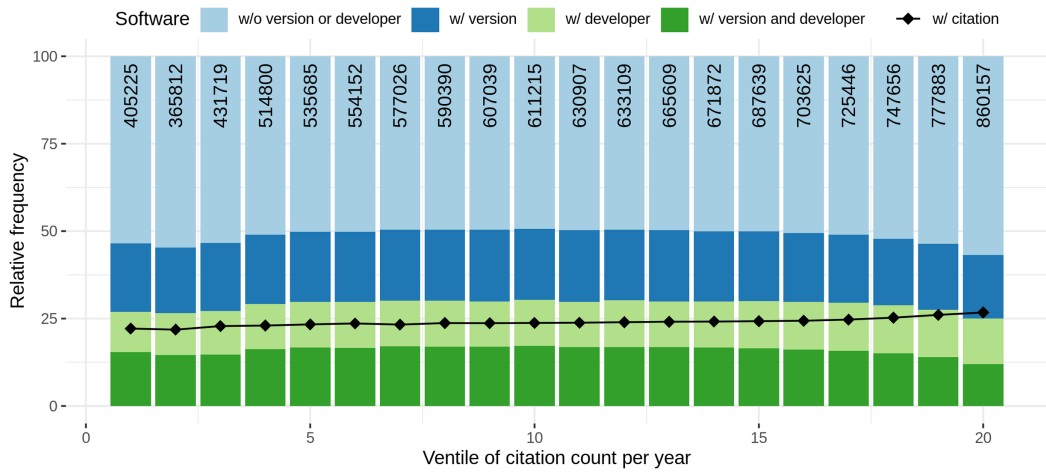

**Figure 14 Distribution of software completeness per ventile of citation count per research domain.** Note that only articles published before 2020 were included to prevent a bias towards lower citation ventiles.

reaching a first local minimum at rank 8 and the absolute minimum at the highest rank. The use of formal citations, in contrast, shows an increase from ~10% at the lowest ranked journals to about 30% at the highest ranked journals.

Considering the citation rank (see Fig. 14), a slight increase in provided additional information could be observed in the first four ventiles and a slight decrease in the last four. No notable difference could be observed for the remaining citation ventiles. For formal citation, a slight increase over the ventiles can be observed, starting with 20% and reaching up to over 25%.

## Types of software mention

For each software mention, the KG contains information about the type of mention and the type of software. The most frequent type of software is Application with 84.49%, followed by Programming Environment with 7.29%, PlugIn with 6.27%, and Operating Systems with 1.95% of the mentions. When looking at the disambiguated software, 88.52% of the software are Applications, 10.74% PlugIns, 0.41% Programming Environments, and 0.33% Operating Systems. With respect to the type of mention, we observed that most mentions of software reflect Usage with 82.31%, whereas 15.09% represent Allusion. Only 2.01% (0.6%) of the software mentions represent Creation (Deposition) statements. Table 11 gives a fine-grained overview of the relation between type of mention and type of software. The mention type Usage is prevalent over all software types and the software type Application over all mention types. Furthermore, we found that the relative frequency for both Creation and Deposition is 0 for Operating Systems and Programming Environments. When looking into domain specific differences, it can be observed that Mathematics (51.4%), Engineering (55.0%), and Computer Science (55.0%) have the lowest share of usage statements, while disciplines such as Energy (90.2%), Materials Science (93.1%), Nursing (93.3%), Dentistry (93.9%), and Veterinary (95.7%) have the highest share of usage statements. The opposite trend can be observed for software

**Table 11 Overview of the relative frequency of software and mention types as well as their combinations over all software mentions.** Note that overall numbers do not necessarily sum to 100 due to rounding issues.

|  | Allusion | Creation | Deposition | Usage | Overall |
|---|---|---|---|---|---|
| Application | 13.95 | 1.80 | 0.56 | 68.14 | 84.49 |
| OperatingSystem | 0.26 | 0.00 | 0.00 | 1.69 | 1.95 |
| PlugIn | 0.47 | 0.17 | 0.04 | 5.58 | 6.27 |
| ProgrammingEnvironment | 0.40 | 0.00 | 0.00 | 6.88 | 7.29 |
| Overall | 15.09 | 2.01 | 0.60 | 82.31 | 100.00 |

**Table 12 Most frequent host software, i.e., mentioned together with a PlugIn, in combination with the most frequently used PlugIns for each of them.** # PlugIn, distinct, disambiguated PlugIns; # Mention, overall PlugIn mentions.

| Software | # PlugIn | # Mention | Top 5 PlugIn incl. % of mentions |
|---|---|---|---|
| R | 19,442 | 220,750 | Bioconductor (4.79%), ggplot 2 (3.63%), lme 4 (3.21%), vegan (3.09%), DESeq 2 (2.52%) |
| MATLAB | 4,442 | 18,616 | Psychophysics Toolbox (6.75%), Psychtoolbox (6.03%), Statistics Toolbox (3.92%), Image Processing Toolbox (2.87%), Neural Network Toolbox (1.43%) |
| Python | 3,157 | 11,688 | scikit - learn (12.55%), SciPy (4.10%), TensorFlow (3.75%), Network (2.37%), scipy (2.30%) |
| python | 1,449 | 3,533 | scikit - learn (10.44%), scipy (3.85%), sklearn (2.83%), matplotlib (2.52%), HTSeq (2.12%) |
| ImageJ | 1,286 | 10,761 | Fiji (44.10%), NeuronJ (3.00%), Cell Counter (2.77%), MTrackJ (2.11%), Analyze Particles (1.64%) |
| Stata | 809 | 2,190 | metan (5.94%), runmlwin (3.06%), mvmeta (2.33%), Image Composite Editor (1.87%), metareg (1.78%) |
| Perl | 774 | 1,176 | MISA (3.74%), speaks - NONMEM (2.64%), Bioconductor (2.21%), NONMEM (1.36%), Shell (1.11%) |
| Excel | 644 | 1,946 | XLSTAT (17.83%), nSolver (3.96%), Microsatellite Toolkit (2.77%), Analysis ToolPak (2.00%), @ Risk (1.95%) |
| Cytoscape | 553 | 5,902 | ClueGO (13.62%), MCODE (13.00%), BiNGO (7.66%), NetworkAnalyzer (7.56%), Enrichment Map (5.71%) |
| SPM | 521 | 2,671 | DARTEL (20.10%), MarsBaR (9.55%), Marsbar (2.92%), CONN (2.62%), DPARSF (2.55%) |

Allusion statements, starting with Veterinary (3.9%) and ending with Mathematics (40.7%). Software Creation statements have the highest proportion in Mathematics with 6% of all mentions, followed by Engineering (5.4%) and Computer Science (5.4%). In Dentistry and Veterinary only 0.3% and 0.4% are Creation statements. Decision Sciences (1.8%), Mathematics (1.7%) and Computer Science (1.5%) have the highest share of deposition statements, in contrast, less than 0.1% of software mentions are Deposition statements in Materials Science and Veterinary.

The software type PlugIn plays a special role, as it can only be used together with another software, requiring the mention of both, the host software and the PlugIn. Table 12 lists the top 10 host software together with the number of PlugIns identified for this software, the overall amout of PlugIn mentions for the software and its top 5 PlugIns. The Programming Environment R was found to be by far the software with most PlugIns (19,442 distinct PlugIns), followed by Matlab and Python. As most frequently mentioned PlugIns for R we found Bioconductor, ggplot2, lme4, vegan, and DESeq2 which together account for 17.2% of all R PlugIn mentions. Note that the two different spellings *Python* and *python* were not linked together but reflect a similar distribution of PlugIns.

**Table 13 Top 10 most frequent URLs accompanying software deposition and usage statements together with their absolute and relative frequencies.**

| Deposition | | | Usage | | |
|---|---|---|---|---|---|
| URL | Absolute | Relative | URL | Absolute | Relative |
| github.com | 8,602 | 13.93 | github.com | 18,918 | 3.90 |
| journals.plos.org | 5,926 | 9.60 | ncbi.nlm.nih.gov | 16,832 | 3.47 |
| sourceforge.net | 918 | 1.49 | r-project.org | 13,176 | 2.71 |
| cran.r-project.org | 673 | 1.09 | pacev2.apexcovantage.com | 10,504 | 2.16 |
| bioconductor.org | 651 | 1.05 | ebi.ac.uk | 9,850 | 2.03 |
| ebi.ac.uk | 478 | 0.77 | blast.ncbi.nlm.nih.gov | 8,797 | 1.81 |
| ncbi.nlm.nih.gov | 454 | 0.74 | cbs.dtu.dk | 6,539 | 1.35 |
| bitbucket.org | 423 | 0.69 | fil.ion.ucl.ac.uk | 6,439 | 1.33 |
| code.google.com | 353 | 0.57 | cran.r-project.org | 6,015 | 1.24 |
| string-db.org | 204 | 0.33 | targetscan.org | 5,738 | 1.18 |

## Software creation and deposition

Software Usage and Deposition statements are often accompanied by URLs to provide a location to access a software and make it findable for the scientific community. Table 13 shows the most common domains of URLs mentioned in combination with software Usage and software Depositions. Usage domains (right) correspond to specific software, for instance, blast.ncbi.nlm.nih.gov or r-project.org for BLAST and R, but also to software repositories such as GitHub and specific software package repositories such as CRAN cran.r-project.org. For Depositions (left) we found that most of the URLs point to source code and software repositories. GitHub (github.com) is the most frequent domain with 14% of overall URLs, but other public repositories such as SourceForge, BitBucket or Google Code are present as well as repositories focused on packages such as CRAN and BioConductor.

Another aspect of recognizing software Creation and Deposition statements in articles is that it allows to identify journals that are most frequently used for the description of new software. By analyzing the relative number of articles per journal that contain either a software Creation or Deposition statement, we were able to find the most active journals when it comes to software description. Considering only journals with at least 10 articles, we found that with 90% of the articles containing software Creation statements, the *Proceedings of the VLDB Endowment* is ranked highest among journals introducing software. It is followed by the journal *Source Code for Biology and Medicine* with 84.4%, *Database: the journal of biological databases and curation* with 74.4%, and the *Journal of Open Research Software* and *Bioinformatics* with 72.7% and 70.8%, respectively. The journal *Source Code for Biology and Medicine* contains the most articles with software Deposition statements (64.7%), followed by the *Journal of Open Research Software* and *Neuroinformatics* with 54.5% and 47.1%, respectively. From 15,388 journals, only 3,622 journals contain at least one article with either Creation or Deposition statements.

## DISCUSSION

### Reliable method for software mention extraction

Information extraction is based on reliable ground truth data from SoMeSci (IRR $F = 0.93$, $\kappa = 0.82$). In combination with state-of-the-art language models for scientific articles such as SciBERT, we achieve state-of-the-art performance for software mention extraction in scientific articles. Regarding software recognition, the SoMeSci baseline was outperformed by a notable margin raising performance to $F = 0.88$ from $F = 0.83$ by 5 pp. This also represents an increase over the previous automatic approaches by *Pan et al. (2015)* with $F = 0.58$, *Duck et al. (2016)* with $F = 0.67$, *Lopez et al. (2021)* with $F = 0.74$ and *Schindler, Zapilko & Krüger (2020)* with $F = 0.82$, however, as prior work was based on different data bases, the results are not directly comparable. With respect to the chosen NER architecture, SciBERT achieved superior recognition rates when compared with Bi-LSTM based models, illustrating the effectiveness of SciBERT for mining scholarly documents. Interestingly, *Lopez et al. (2021)* report notable lower performances for both architectures, which we believe results from the less reliable input including PDF conversion artifacts and ground truth annotation.

Regarding the identification of additional information ($F = 0.89$, baseline $F = 0.85$) as well as software type ($F = 0.80$, SoMeSci baseline $F = 0.78$) and mention type ($F = 0.81$, SoMeSci baseline $F = 0.74$), we achieve better performance than baseline results, especially considering that the reported results already take error propagation into account, which is not the case for baseline results. We also achieve better performance for Version, Developer, and URL as reported for the Softcite corpus (*Du et al., 2021*; *Lopez et al., 2021*), however, these results cannot be directly compared due to different training and test data. Moreover, the study presented here is the first that classifies software mentions according to both, software and mention type. However, we found that software type PlugIn and mention type Allusion were extracted with lower performance as other types. In both cases the lower performance is mainly due to confusion with another class (Application and Usage) with corresponding higher prior probability but a difficult to distinguish context. This is consistent with the results of the manual annotation performed for SoMeSci (*Schindler et al., 2021b*), where annotation IRR was also found to be lowest for these classes.

RE and disambiguation for software mentions has, to the best of our knowledge, not been evaluated as part of any scientific investigation of software mentions in scholarly publications besides the SoMeSci baseline. We improve RE performance by 6 pp from $F = 0.88$ to $F = 0.94$. RE performs well for additional information related to software, but it is challenging ($F = 0.71$–$0.78$) to predict relations between software entities such as the *plugin-of* relation. This was expected since, by definition, additional information is always related to another entity while two software entities are not necessarily related to each other. Consequently, relations between software are rare compared to the overall number of software mentions. It should also be noted that both, baseline and our reported results, do not take error propagation from entity recognition to RE into account. Overall, we achieved superior recognition rates compared to previous, automatic, large scale analyses

of software mentions in scholarly publications and conclude, thus, that software mentions and additional information extracted by our pipeline are more reliable.

With respect to disambiguation, it has to be noted that previous large scale analyses did not consider spelling variations but only summarized software mentions with equal (or similar) spellings. As this is the first study to use disambiguation methods for software mentions, comparison to state-of-the-art results is not possible. However, the disambiguation baseline performance for SoMeSci of $F = 0.97$ was matched, while considering a much denser feature space. In small data sets, only few spelling variations (and other features used for disambiguation) of software exist; this number increases with the size of the data set. This means that finding reliable boundaries between different software gets harder with increasing size of the data set as rare spelling variations (and other features) of software with similar names tend to overlap stronger with increasing amounts of data. In our case, the training data set contains 3,756 software mentions from 637 different software while information extraction resulted in almost 12 M software mentions. To recreate this effect for the training data, we included a large set of augmented, fictional software names. With respect to evaluation, the negative effect of increasing sample size on the ability of finding reliable boundaries between different software prevents the transfer of quality statement from training to inference dataset. To counteract this effect, we included the manually disambiguated training data in the inference dataset, determined the clustering threshold, and evaluated the quality based on those samples. Same as for RE, it should be noted that error propagation from the previous information extraction steps influence disambiguation performance. The additional augmented samples simulate the effect of false positives, but we cannot estimate to what extent they are successful at suppressing resulting errors. False negative samples do in practice directly influence linking quality.

Due to computational and space complexity we chose a single linkage-based clustering, which is known for semantic drift away from cluster means, but enables an efficient implementation when distance between all pairs is pre-computed and sorted up to a given threshold. Average linkage would have required to re-compute the average distance of all clusters in each step. An initial evaluation showed only marginal differences between single and average linking based clustering for disambiguation, which seemed sufficient for the task at hand. Overall, disambiguation provides reasonable results; 440 different spellings for SPSS[1], have, for instance, been discovered. For the different spellings python and Python (see Table 12), however, no common cluster could be determined. While this clearly represents an error when considering the string only, our machine learning based distance additionally considers the context and accompanying entities such as developer and URL. We believe that the reason here is the low number of spelling variations that prevent the semantic drift to counteract misleading linking probabilities. In consequence, this would mean that more frequent software (with many different contexts and spelling variations) are more likely to be disambiguated than less frequent software (with fewer contexts and spelling variations), which may have had a reinforcing effect on the power law distribution of mentioned software. For software with more frequent spelling and context variations, in contrast, this might result in more false positives and

[1] All 440 different spellings of SPSS have manually been validated.

thus overestimate the software use. For Excel, 54 spelling variations were found that represent 152 K mentions. From those only about 150 K mentions (from 13 different spellings) can be considered as correctly classified. The remaining mentions contain software such as Firefox (1,162, 0.7%) or F (185, 0.1%). A similar phenomenon could be observed for Stata. While disambiguation performance is satisfactory the algorithm can be improved in future work, for instance, by including information on PlugIns provided with software names after an initial disambiguation of the PlugIn names. However, this would lead to higher run-time requirements because a higher number of mention contexts needs to be considered to cover rare features such as PlugIns. The number is currently limited to $n = 6$.

## SoftwareKG: knowledge graph of software mentions

SoftwareKG represents the largest dataset of software mentions and related metadata in scholarly publication. It contains 11.8 M software mentions of over 605 K different software automatically extracted from more than 3 M Open Access articles from PMC. Moreover information from PubMedKG was integrated to allow bibliometric analyses. The KG was created by re-using established vocabularies for data representation, such as schema.org, BIBO, and DCT and is available as JSON-LD under Creative Commons Attribution at Zenodo (*Schindler et al., 2021a*). The published version of the KG only contains information available under open licenses. As this is not the case for most of the bibliometric data, those parts where excluded from publication.

SoftwareKG consists of over 300 M triples describing the properties and relations between more than 55 M resources. A summary of the properties of SoftwareKG is given in Table 14. In SoftwareKG, we employ frequency-based confidence values to provide a transparent way to analyse errors that originate from information extraction or author spelling variations. For names, developers as well as software type and other information we included those confidence values in the reification statements to allow further analyses.

SoftwareKG facilitates the large-scale analysis of software mentions in scholarly publications and allows to give insights into the role of software in science. A tutorial to recreate all tables and figures from the KG is included in the Supplemental Material (https://github.com/f-krueger/SoftwareKG-PMC-Analysis). This article contains first analyses and sketches the potential for more elaborate studies. This includes the creation of an impact measure for scientific software but also to provide a software mapping for science in general such as swMath (*Greuel & Sperber, 2014*) for Mathematics.

## The role of software in science/PMC

### Software mentions

With an average of 3.67 software mentions per article, our result confirms previous studies, ranging from 2.6 (*Schindler et al., 2021b*) to 3.2 (*Howison & Bullard, 2016*) to 5.5 (*Duck et al., 2016*) in different subsets of PMC. With over 605 K, the number of different software from over 11.8 M mentions is high, given that 3.2 M articles were investigated. This number probably overestimates the actual number of software used in science, due to errors from information extraction and disambiguation. The distribution of software

**Table 14 Statistics of SoftwareKG. Left: general KG properties. Right: frequencies of resources per type.**

| Property | Frequency |
| --- | --- |
| Triples | 301,825,757 |
| Resources | 55,953,270 |
| Distinct Types | 12 |
| Distinct Properties | 47 |
| Reification Statements | 2,042,076 |

| Type | Frequency |
| --- | --- |
| nif:String | 22,066,759 |
| schema:Person | 20,373,227 |
| schema:Organization | 7,063,708 |
| schema:ScholarlyArticle | 3,215,346 |
| rdf:Statement | 2,042,076 |
| irao:Software | 605,352 |
| skg:SoftwareVersion | 380,234 |
| skg:JournalInformation | 134,369 |
| bibo:Journal | 15,338 |
| dct:LicenseDocument | 4,748 |
| skos:Concept | 303 |
| skos:ConceptScheme | 27 |

mentions per software shows that only few software are used in a large number of articles and thus play a major role in science. This distribution partly confirms the general trend but shows even higher skewness as previously reported statements about the distribution of software mentions in scholarly articles (*Pan et al., 2015*; *Duck et al., 2016*). This amplified trend could be the result of software name disambiguation which was not applied in previous studies and highlights the importance of considering all spelling variations for software usage analysis. The most frequent software (7 from the top 10) are mainly used for statistical data analysis. A closer look at the domain specific distribution of the top 10 software revealed domain specific differences as it characterizes all of the analysed domains uniquely. The software SHELXL and SAINT, for instance, are most frequently but exclusively used in Chemistry, Materials and Physics, whereas Excel is frequently used in almost all other research domains except for them. Overall, an increased role of applications that can be controlled *via* scripts rather than point and click software can be observed. *Schindler, Zapilko & Krüger (2020)* reported that the Programming Environment R superseded SPSS in an excerpt of articles in PLoS One from 2017. While a similar trend can be seen here, the particular ranks did not change yet, see Fig. 15. While the usage of SPSS, Excel, and SAS remained constant over the last 5 years at the relative level, usages of R and Python increased. Considering articles from PLoS One only, R replaced SPSS at the top position, which confirms the result and suggests journal specific software preferences.

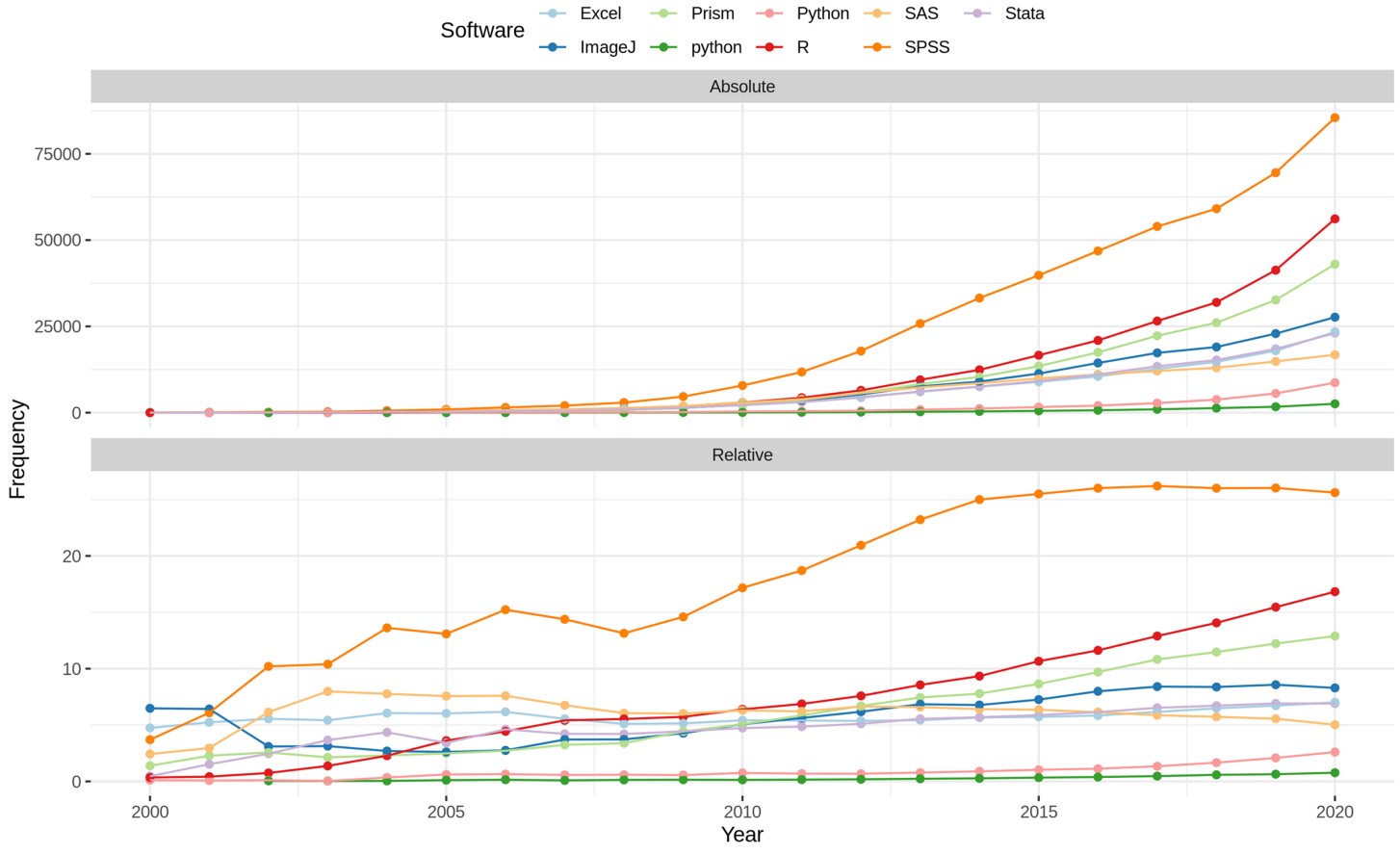

**Figure 15 Relative and absolute amount of articles per year mentioning the top statistical software.**

### Article level statistics

The importance of software increased in recent years for both, the actual investigation as well as the reporting within the scholarly publication. This is suggested by the increasing trend of including software mentions into the textual description and the increasing number of different software per article. A reason could be the growing complexity of data driven analyses requiring more software to be employed, coupled with a high awareness about transparency and reproducibility in general. The positive correlation between journal rank and number of different software supports this by suggesting strong rigor in the description of the analysis[2]. The positive correlation of the number of software and the number of citations per article indicates a growing appreciation of the traceability of the described research processes. The observed domain specific difference in software usage could reflect the role of data in those domains. While in Arts and Humanities and Economics only few articles mention only few software suggesting an important role of manual data analysis, in Mathematics, Computer Science, Decision Science, and Agricultural and Biological Science many articles mention multiple different software indicating automatic and complex data collection and analyses.

[2] When interpreting the journal rank as an indicator of journal quality and thus for review quality.

### Software citation completeness

Citation completeness has not improved over recent years, that is, the information provided to identify the particular software is not provided to full extent. This suggests a lack of awareness about the necessity to understand and reproduce research processes and its requirement for identifying particular software versions. When comparing formal citation, *i.e.*, providing a formal literature reference, with in-text mention, we found contrary trends both over time and across categories. Here, we consider in-text citations as complete when version and developer are included, while formal citations are always considered sufficient. Following this definition, only about 38% of the articles in 2020 allow the unique identification of the used software based on the provided information. Software usages in the technical domains use formal citation more frequently in contrast to research domains related to medicine. We believe the reason to be that the latter more frequently employ other materials and devices and adapt the same citation style for all research objects other than scholarly publications. With respect to journal rank, we see a growing trend in the usage of formal software citation with rising rank and a contrary trend in the completeness of in-text software mention. We believe that this supports the statement of increasing rigor in scholarly review and the request for more traceable descriptions in higher quality journals. This is also supported by the slight increase of formal citation frequency and contrary decrease of citation completeness for rising citation count. In summary, our results indicate that software citation standards, as suggested by *Katz et al. (2021)* or *Smith, Katz & Niemeyer (2016)* have not been adequately adapted in scholarly publications, yet.

### Types of software mention

Each software mention is classified according to mention type indicating the reason why the software was mentioned within the scholarly article and software type providing information about the particular kind of software. Analyses of disciplinary differences showed that most software is created and published by scientists from the technical domains (Mathematics, Engineering and Computer Science). This reflects the domains with the highest interest in automating complex calculations combined with the programming knowledge to implement new suited software applications. Further, software allusions without actual usage are also more common within scholarly publications from those disciplines indicating more description, discussion and comparison between software entities. On the other hand there are disciplines which mostly reuse existing scientific software such as Material Science, Nursing, Dentistry, and Veterinary.

When looking at the different kinds of software, we found an increasing trend of using PlugIns over recent years, see supplementary Fig. A1. The relative frequency of Application mentions, in contrast, declined. We see this as an indicator toward the usage and extension of established software frameworks. With respect to the host software, we found a notable overlap in the most frequently used software (Table 10) and the most important host software (Table 12). This includes the Programming Environments R and Matlab, but also the Applications ImageJ, Stata, and Excel. More than 19 K PlugIns were found for the Programming Environment R making it the most important host

software for scientific investigations. While this number seems high at first glance, inspecting the two most important package repositories, CRAN (https://cran.r-project.org/) and Bioconductor (https://www.bioconductor.org/) with 18,312 and 2,042 unique packages[3] indicates these results to be plausible. However, the comparably low FScores for the identification of PlugIns might have resulted in an overestimation of less frequent PlugIns. This growing interest in the Programming Environment R and its package universe was previously investigated (*Li, Yan & Feng, 2017*; *Li & Yan, 2018*), some results of which are confirmed here. In particular, we see an overlap for the most frequent R packages.

### Software creation and deposition

By analyzing the mention types Creation and Deposition, we were able to identify the most important targets for the publication of software. On the one hand this includes web services such as GitHub for general purpose software and CRAN for R packages, on the other hand software journals. Specifically designed repositories to host and assign Digital Object Identifiers (DOI) to scientific research data such as Zenodo are not commonly used for publishing scientific software with a share of <1% of depositions. While this allows to provide researchers with recommendations on where to publish their software and/or the corresponding description, it also enables the search for software. Moreover, the identification of Creation and Deposition allows to track the scientific software landscape with low latency. It has to be noted that the second most frequent deposition URL is the result of a false software mention detection and its propagation.

### Summary

The importance of software in science has been growing in recent years, in both relative and absolute numbers. The awareness for providing all necessary information to enable the identification of the particular software by others, in contrast, remains unchanged. Software citation principles have not been adapted yet in scholarly publications. However, articles in higher ranked journals tend to more formal software citations instead of in-text citations, which reflects recent software citation recommendations (*Katz et al., 2021*). Articles in lower ranked journals provide more complete in-text citations, *i.e.*, Version and Developer. We identified domain specific software citation habits: Medicine related domains prefer in-text citation, while technical domains tend to more formal citations. Domain independent as well as domain specific software is used across most research domains, the top 10 of which represent domain specific characteristics. Most software mentioned in scholarly articles are software for statistical analysis, such as SPSS, R, and Prism. Interestingly, we identified an increased interest in the usage of PlugIns, which allow the problem specific extension of general purpose software. The most important representatives of them are the Programming Environments R, MATLAB, and Python. Finally, we confirmed GitHub as a central repository for scientific software, for both publication and re-use, as previously assumed (*Russell et al., 2018*).

---

[3] Package counts for CRAN and Bioconductor were retrieved on October 4th, 2021.

## Limitations of the study

The study presented in this article involves complex data processing and information extraction steps, where each of these is subject to limitations that are discussed in the following.

The articles in SoftwareKG cover a broad range of scientific disciplines, however, the selection of PMC as primary data source implies a bias towards Medicine. While the training data set SoMeSci itself is also taken from PMC, the selection might introduce domain specific biases. Therefore, the trends reported here, might be different if we look at a domain such as computer science in general. For instance, BLAST is the second most used software in computer science in our set, which would likely not be true when looking at computer science in general, as this software is primarily used in Bioinformatics. Another bias in the selection of articles is towards open access, as all article are from the PMC Open Access subset. Researchers choosing to publish under open access might also be supporters of open data movement and, therefore, have a better awareness for attributing other open work such as software.

For information extraction and disambiguation, we found high performance for all employed machine learning methods. However, it is important to consider error propagation between them. The given evaluation for software and mention type classification does take error propagation into account, but the results for RE and entity disambiguation do not. Therefore, the $F = 0.94$ performance for RE might overestimate the true performance as it relies on results of $F = 0.885$ entity recognition. For disambiguation we model the effects of false positive entities by data augmentation, but it is hardly possible to tell if this completely suppresses their effect and false negatives do directly influence disambiguation performance. Moreover, evaluation for disambiguation has proven to be challenging and the gold standard dataset alone is no good predictor for performance on large scale entity disambiguation. We, therefore, adjusted our evaluation method to take the large scale data into account, but note that further systematic evaluation is required for entity disambiguation.

Our analyses regarding domains, journal rank and citation count rely on external data and are, thus, influenced by their quality. For instance, only 86.7% of our data was covered by Scimago data on domains and journal rank. Regarding the external citation data we assumed completeness but in case article citations were missing from the list they were not included in the computed citation count. In case one article would miss completely it would be counted with a citation count of 0.

An analysis of the article types (*skg:documentType*) contained in SoftwareKG showed that aside the largest group of research articles it also covers review articles and abstracts, but also case reports or letters and several other categories. For each of the groups we did find publications that cite software, but the prior probability for software mentions across article types differs. Therefore, it is important to note that the reported results are not specific to only research articles but to the distribution of scientific publications indexed in the PMC OA subset. The information about the article type, however, is included in the KG enabling others to analyse these effects.

## CONCLUSION

In this article we presented the largest analysis of software usage in scholarly publications over the longest duration covering articles between 1990 and 2021[4]. Software mentions were identified by automatic information extraction covering NER for software and associated information, software and mention type classification, and RE between software and additional information. Moreover, in difference to previous studies, software names were automatically disambiguated to allow reasoning about software usage even when the same software is referred to by different names. The analysis covers 3.2 M articles, mentioning a total of 11.8 M software.

From the extracted information, we created SoftwareKG, the largest KG describing software mentions in scholarly publications. The KG was created by re-using existing vocabularies and published under an Open Access license to support further research on the role of software in science. SoftwareKG consists of over 300 M triples and contains information about software, accompanying information as well as information about articles, journals, authors, and publishers.

We performed a large-scale analysis on SoftwareKG with respect to publication date, article domain, journal rank and article citation count in order to identify differences and trends in software mention. Overall, the results show that software usage has increased over the course of the last 10 years, but we found no change in citation completeness during this time frame. This leads us to believe that there is still a lack in awareness for software citation in science, even so software citation standards have been available and promoted since 2016, *e.g.*, by *Smith, Katz & Niemeyer (2016)*.

We also identified a trend towards using extendable software architectures instead of stand-alone software, especially in combination with the Programming Environment R. Overall, their design allows an easy extension and offers high flexibility. Especially adding functionality and publishing new packages or PlugIns is facilitated. We could also show this trend by analyzing which infrastructure is used by scientists to publish their software, with GitHub playing a central role, but CRAN and Bioconductor being especially important in combination with R.

In general, we show that there are many domain specific peculiarities in software usage. We showed that the amount of software usage as well as the most used software per domain and their application purpose varies significantly. Domain specific citation habits are also reflected in preferences to formal and informal software citation, ranging from 5–40% in formal citation contrasting to 1–35% software citation completeness with opposing trends.

Overall, we believe that SoftwareKG provides a valuable data source for further investigations about the role of software in science. One finding that should be further investigated is, for instance, the influence of journal rank on formal citation of software usage. The trend could, for instance, be explained by higher review quality and journal policies enforcing better software citation. Further insights could allow to give better recommendations for journals to encourage software citations habits.

In future work SoftwareKG can build the basis to further explore software usage in science, for instance, as a mapping for available software and newly established software.

[4] SoftwareKG actually contains articles from before 1990, but we restricted most analyses to time between 1990 and 2021.

It can also be used to track software usage and establish software impact measures. Furthermore, the investigation of formal software citations should be extended to include the citation targets. Currently, formal citations are recognized, but not further analysed. In the future we need to include a distinction between software citation and software article citation and model citation completeness within formal citations.

## Software and Data

We implemented all machine learning models for information extraction in Python 3.9.5 (*Van Rossum & Drake, 2009*), utilizing the following packages: PyTorch 1.9.0 (*Paszke et al., 2019*) for deep learning models, Huggingface transformers 4.9.1 (*Wolf et al., 2020*) to load and fine-tune pre-trained BERT models, Gensim 4.0.1 (*Řehůřek & Sojka, 2010*) for pre-training word embeddings, scikit-learn 0.24.2 (*Pedregosa et al., 2011*) for implementation of RE models, articlenizer R-14.06.2021 (*Schindler, Zapilko & Krüger, 2020*, https://github.com/dave-s477/articlenizer) for preprocessing of scientific articles, and NLTK 3.6.2 (*Loper & Bird, 2002*) for feature extraction. Moreover, to extract JATS XML meta data we used lxml 4.6.3 (*Behnel, Faasen & Bicking, 2005*) and for knowledge graph construction rdflib 6.0.0 (RDFLib Team https://github.com/RDFLib/rdflib). For statistical analysis and generation of figures we used R 4.1.1 (*R Core Team, 2021*), utilizing tidy verse 1.3.1 (*Wickham et al., 2019*) for data processing and plotting and SPARQL 1.16 (*van Hage et al., 2013*) to access the KG interface. To setup a SPARQL endpoint for SoftwareKG we used OpenLink Virtuoso Open Source Edition 07.20.3229 (*OpenLink, 2021*), available as docker from https://hub.docker.com/r/tenforce/virtuoso/.

Source code for construction and analysis is published on GitHub at https://github.com/f-krueger/SoftwareKG-PMC-Analysis and the data for SoftwareKG (*Schindler et al., 2021a*) itself is available on Zenodo *via* https://doi.org/10.5281/zenodo.5553737. To facilitate reproducibility of our data analysis, a docker file including a suited R environment to execute analyses on SoftwareKG is included.

## ABBREVIATIONS

| | |
|---|---|
| **KG** | Knowledge Graph |
| **IRR** | Inter-Rater Reliability |
| **PMC** | PubMed Central |
| **pp** | percentage points |
| **JATS** | Journal Article Tag Suite |
| **NER** | Named Entity Recognition |
| **RE** | Relation Extraction |

### Funding

This work was financially supported by the Deutsche Forschungsgemeinschaft (DFG, German Research Foundation) as part of the projects SFB 1270/2 (grant: 299150580) and ScienceLinker (grant: 404417453). Parts of the computation were done by using a

computer cluster funded by DFG (grant: 440623123). The funders had no role in study design, data collection and analysis, decision to publish, or preparation of the manuscript.

### Grant Disclosures
The following grant information was disclosed by the authors:
Deutsche Forschungsgemeinschaft (DFG, German Research Foundation) SFB 1270/2: 299150580.
ScienceLinker: 404417453.
DFG: 440623123.

### Competing Interests
The authors declare that they have no competing interests.

### Author Contributions
- David Schindler conceived and designed the experiments, performed the experiments, analyzed the data, performed the computation work, prepared figures and/or tables, authored or reviewed drafts of the paper, and approved the final draft.
- Felix Bensmann performed the experiments, performed the computation work, prepared figures and/or tables, authored or reviewed drafts of the paper, and approved the final draft.
- Stefan Dietze conceived and designed the experiments, authored or reviewed drafts of the paper, and approved the final draft.
- Frank Krüger conceived and designed the experiments, performed the experiments, analyzed the data, performed the computation work, prepared figures and/or tables, authored or reviewed drafts of the paper, and approved the final draft.

### Data Availability
The source code is available at GitHub: https://github.com/f-krueger/SoftwareKG-PMC-Analysis.

The data is available at Zenodo: Schindler, David, Bensmann, Felix, Dietze, Stefan, & Krüger, Frank. (2021). SoftwareKG-PMC (0.2) [Data set]. Zenodo. https://doi.org/10.5281/zenodo.5713973.

### Supplemental Information
Supplemental information for this article can be found online at http://dx.doi.org/10.7717/peerj-cs.835#supplemental-information.

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
