# Peer review of "The role of software in science: a knowledge graph-based analysis of software mentions in PubMed Central"

_PeerJ Computer Science, doi:10.7717/peerj-cs.835_

## Round 0.1 · original submission · Minor Revisions

While finding your paper interesting and worthy of publication, the referees and I feel that more work could be done before the paper is published. My decision is therefore to provisionally accept your paper, subject to minor revisions.

·

Basic reporting

This is an excellent paper. The scope and purpose of this paper was well documented. The literature review is thorough and reflective. The selected data was expansive and appropriate for the purpose of the proposed research. Methods and data materials were made available to the general public.

Experimental design

Software recognition is a complex task. The reviewer worked on several projects that intended to extract software entities in his past research and understand the challenges involved. Thus, I am very appreciative of the the method proposed and the large-scale data analytics reported in this research.

The method is capable of not only identifying software entities but also several associated metadata which is quite valuable. The interpretation of results were insightful as they included discipline-, journal-, and timeline-based analysis. The improved methods were compared with baseline methods and it was clear a better performance was attained.

A few minor comments:
1. It may be helpful to define what software is. Based on the results, OS, environments, plugins, and applications were all included as software. It is useful to draw theorization from STS/science of science to operationalize software for purpose of gaining meaningful results.
2. It is unclear what negative sentences meant in Table 2.
3. It is unclear what is n is Tables 7 and 8.

Validity of the findings

The figures reported in the results section were informative. Journal- and discipline-level differences can be readily discerned. I wonder if there is any artifact in Table 8 in which late 1990s and early 2000s seemed to have more software mentions with developers than any other periods.

Additional comments

The softwareKG is very valuable and it is likely to garner attention by fellow researchers to plan further research in this area.

·

Basic reporting

In this paper, a data extraction process is given to extract data about scientific research which employs software. The paper is well-written and does not require any modification in terms of language use. An up-to-date background and literature review/related work are provided. The paper is well-organized and supported with appropriate figures and tables.

Experimental design

The research question is defined and how this research fills the identified gap is clearly explained. Three specific contributions (which implicitly define the RQs) are formulated as (1) A large-scale analysis of software usage, (2) A comprehensive knowledge graph of software citations, and (3) Robust supervisor information extracting models for disambiguating software mentions and related knowledge.

Methods to implement the aforementioned questions/contributions are properly explained and the relevant data sets were defined and cited. The data model for the Knowledge Graph was put forward and the information extraction process using the machine learning algorithms was investigated in detail. Performance of the BERT and ML,sw,opt models are compared in terms of Precision, Recall, and FScroe parameters.

Validity of the findings

The results for the three contributions were discussed along with their limitations. All underlying data have been provided and appear to be robust and controlled. Conclusions are well-stated and the future work is included.

·

Basic reporting

This work describes an approach for detecting software mentions in literature by combining named entity recognition, relation extraction and entity disambiguation techniques. As a result, the authors create a dataset of thousands of software mentions in papers, represented as a knowledge graph, and using reification to indicate the confidence of the results. The authors then proceed to analyze the role of scientific software in different domains; sharing insights on how software is becoming a first class citizen in most disciplines. Many software creators would be happy to see how thanks to this work they can assess how influential their software is.

The paper is well written and easy to follow. I think this work is timely and highly relevant for PeerJ and the Open Science community. The paper is thorough, and does not omit important implementation details such as the hyperparameters used or the pointers to the tools used for the conversion into annotations for the models, which always take significant time to piece together. Therefore, I think the paper would be a nice addition to the journal, and I look forward to reusing some of the outcomes made available by the authors.

I have some comments and questions that would be great to address/clarify in the final version of the paper (my "minor revision" is really minor). I add them in the "additional comments" field

Experimental design

Robust and thorough experiments. Nice discussion of the results, highly related to the journal. Some questions to be addressed in the "Additional comments" field below

Validity of the findings

The findings and analysis are highly interesting, build on prior work and improve the state of the art. I have found a few errors regarding the vocabularies used, please see "Additional comments" below

Additional comments

- I found that the skg namespace (http://data.gesis.org/softwarekg2/) returns a 404, which will harm the reusability of the endpoint (I see examples online, but the data model documentation is what is critical for performing queries). I also see that some properties use http://data.gesis.org/softwarekg/software, which also returns a 404.

- The authors present baseline results in the same table as the results of the paper. However, in the text, they insist that direct comparison is not appropriate. Then why adding tables comparing both approaches? I think it misleads the reader.

- Having the ability to find whether software is created or repurposed in a publication is great. Are there areas where this happens more than others? I think I have not seen this part in the discussion.

- I am surprised not to see Zenodo as a common archive for depositing software. Are authors (in general) not depositing their releases there, or it something out of the scope for this paper? I know that it's a common practice for researchers to point to their Zenodo releases through the GitHub release integration.

- The relationship extraction between software is deemed as challenging in the paper. However, I have not found why. Is there an explanation for this?

- I find curious some of the choices by the authors. For example, programming languages are marked as Software, and packages in those programming languages are plugins. Is this representation easier for the training part?
- Also, I wonder if for the clustering/disambiguation, the plugins could play a part in the similarity metric. At least in that case the python/Python separation would be fixed.

- I imagine the authors have tweaked the BERT models used in their evaluation. However, I have not found any of the tweaked models in any of the links provided by the authors. Where can the models be found?

Minor:

- The github repositories provided with the resources are not tagged (do not have releases), and therefore may be difficult to rerun the experiments if one changes.

- Typo: Page 6: Objective of this information extraction step -> The objective. Similar thing in line 354.
- Missing "." in line 802.

---

## Round 0.2 · accepted · Accept

We are happy to inform you that your manuscript has been accepted for publication. The reviewers' comments have been addressed. There are several typos to be corrected.

·

Basic reporting

The authors addressed all my concerns.

Experimental design

The authors addressed all my concerns.

Validity of the findings

The authors addressed all my concerns.

·

Basic reporting

All corrections suggested by the other two reviewers are handled with care. No additional comment!

Experimental design

No comment.

Validity of the findings

No comment.

·

Basic reporting

The authors have answered all my questions (reflecting them in the paper when appropriate) and addressed all my minor concerns about the paper. I think this work will be a nice contribution to this journal.

A few typos:
- line 552 has a restricted overleaf link which I think it's not intended there.
- Github should be GitHub

Experimental design

n/a

Validity of the findings

n/a

Additional comments

n/a